# Robust Pareto Set Identification With Contaminated Bandit Feedback

## Abstract

We consider the Pareto set identification (PSI) problem in multi-objective multi-armed bandits (MO-MAB) with contaminated reward observations. At each arm pull, with some fixed probability, the true reward samples are replaced with the samples from an arbitrary contamination distribution chosen by an adversary. We consider $(\alpha, \delta)$-PAC PSI and propose a sample median-based multi-objective adaptive elimination algorithm that returns an $(\alpha, \delta)$-PAC Pareto set upon termination with a sample complexity bound that depends on the contamination probability. As the contamination probability decreases, we recover the well-known sample complexity results in MO-MAB. We compare the proposed algorithm with a mean-based method from MO-MAB literature, as well as an extended version that uses median estimators, on several PSI problems under adversarial corruptions, including review bombing and diabetes management. Our numerical results support our theoretical findings and demonstrate that robust algorithm design is crucial for accurate PSI under contaminated reward observations.

## 1 Introduction

Multi-armed bandit (MAB) problem involves decision making under uncertainty in which a finite amount of resources are allocated between a limited number of options (arms) in order to optimize gain over time (or equally minimize regret). In the classical setting, each arm is associated with a reward distribution that is unknown or only partially known at the time of allocation and the information on distributions increase as more observations are made over time (Thompson, 1933; Robbins, 1952; Lai & Robbins, 1985).

Over the last decades, MAB algorithms have been used in a broad range of applications such as medical treatment allocation (Villar et al., 2015), financial portfolio design (Shen et al., 2015), adaptive routing (Awerbuch & Kleinberg, 2004), cellular coverage optimization (Shen et al., 2018), news article recommendation (Li et al., 2010), and online advertising (Pandey et al., 2007). Due to the security concerns in these applications, adversarial MABs have attracted considerable attention. A variety of adversary models are considered that come with different restrictions on the adversary. One of the widely studied attack model is the attack that has a bounded attack value. Auer et al. (2002b) consider adversarial attacks bounded in value and propose the famous Exp3 algorithm for robust learning, and derive both upper and lower bounds on the regret. However, the asymptotic behaviour of the regret bound approaches to that of a linear bound and hence becomes trivial, as the hardness of the competitor sequence increases. This type of adversarial attack is further studied in Stoltz (2005); Audibert & Bubeck (2009); Bubeck & Cesa-Bianchi (2012). Another popular attack model considered in the literature is the attack model that has a limited budget of attack value. In this model, the total amount of corruption injected in reward samples over all rounds is limited by a certain amount. Two notable works that study this kind of attack are Lykouris et al. (2018) and Gupta et al. (2019). Another adversarial attack model is the bounded probability attack model. In this model, an attack can occur at every round with fixed probability. Unlike the attack models mentioned before, this model does not put any restrictions on the attack value. This attack model is considered in Kapoor et al. (2019); Altschuler et al. (2019); Guan et al. (2020); Mathieu et al. (2024); Mukherjee et al. (2021); Wu et al. (2023), and also in this study. Adversarial attacks are also studied within the context of stochastic linear MAB (Bogunovic et al., 2021) and Gaussian process MAB (Han & Scarlett, 2022).

Table 1: Comparison with the related work.

| Work | Setting | Goal | Adversary | Bound |
|------|---------|------|-----------|-------|
| Auer et al. (2016) | MO-MAB | PSI | Adv. free | Samp. com. |
| Altschuler et al. (2019) | MAB | Best arm id. | Obl.,Presc.,Mal. | Samp. com. |
| Auer et al. (2002b) | MAB | Regret min. | Bounded attack | Cum. reg. |
| Drugan & Nowe (2013) | MO-MAB | Regret min. | Adv. free | Cum. reg. |
| Nika et al. (2020) | MO GP | PSI | Adv. free | Samp. com. |
| Zuluaga et al. (2016) | MO GP | PSI | Adv. free | Samp. com. |
| Kapoor et al. (2019) | MAB | Regret min. | Oblivious | Cum. reg. |
| Guan et al. (2020) | MAB | Regret min. | Prescient like | Cum. reg. |
| Lykouris et al. (2018) | MAB | Regret min. | Prescient like | Cum. reg. |
| **Ours** | MO-MAB | PSI | Obl.,Presc.,Mal. | Samp. com. |

Multi-objective MABs (MO-MABs) are another significant extension of the MAB setting where multiple, possibly conflicting objectives are optimized simultaneously. Unlike the single objective optimization, in multi-objective optimization (MOO) problems, it is not possible to identify a single optimal arm in most of the cases. Therefore, in MOO, the aim is to identify the set of Pareto optimal arms which are not dominated by any other arm (see Section 3 for definition of Pareto optimality).

MO-MABs are extensively studied in the non-adversarial, stochastic settings for regret minimization and Pareto set identification (PSI) problems. Auer et al. (2016) propose an elimination-based adaptive arm sampling algorithm, and derive upper and lower bounds on the sample complexity of successful PSI (Theorems 4 and 17 therein). These bounds are $\sum_i \frac{4320}{\left(\Delta_i^{\epsilon_0}\right)^2} \log\left(\frac{12KD}{\delta \Delta_i^{\epsilon_0}}\right)$ and $\Omega\left(\sum_{i=1}^{K} \frac{1}{\left(\tilde{\Delta}_i^{\epsilon_0}\right)^2} \log\left(\frac{1}{\delta}\right)\right)$ respectively, where $\tilde{\Delta}_i^{\epsilon_0}$ and $\Delta_i^{\epsilon_0}$ are their gaps defined with mean differences of arms as well as their accuracy parameter $\epsilon_0$. In their bounds, $K$ is the number of arms, $D$ is the number of objectives and $\delta$ is the confidence parameter. Drugan & Nowe (2013) investigate MO-MAB from the regret minimization perspective using scalarization based methods. These methods turn the multi-objective problem into a single-objective problem, which can be solved efficiently via well-known single-objective bandit algorithms such as UCB (Auer et al., 2002a). Multi-objective variants of Thompson Sampling and Knowledge Gradient algorithms are investigated in Yahyaa & Manderick (2015) and Yahyaa et al. (2014). Regret minimization in multi-objective contextual bandit problems is studied in Tekin & Turgay (2018) and Turgay et al. (2018).

Another line of work (Zuluaga et al., 2016; Hernández-Lobato et al., 2014; Shah & Ghahramani, 2016; Nika et al., 2020; Campigotto et al., 2014) focuses on PSI with Gaussian process priors and propose acquisition strategies to utilize prior induced dependencies between mean arm rewards.

## 1.1 Contribution and Comparison with Related Works

In the literature, single objective MAB problem is studied under various attack models. In most of the MAB literature, the goal is to identify the arm that corresponds to the reward distribution with the highest first order statistic (mean) (Even-Dar et al., 2002). This is only justified when the attack model is assumed to be bounded in value since mean cannot be estimated from samples contaminated with an attack that has unbounded value. However in many applications, it is more plausible to restrict the probability of occurrence of an attack instead of the attack value. For instance, consider review bombing, an internet phenomenon where some accounts post negative reviews or ratings for a product, service, or content as a form of protest or manipulation. In this case, the bounded attack probability represents the fraction of adversarial users in the system. As shown in the single objective adversarial MAB studies that consider bounded probability attack model, the median is a robust measure against the unbounded attacks (Altschuler et al., 2019). Altschuler et al. (2019) work on this setting and provide upper and lower bounds as $\mathcal{O}\left(\sum_{i \neq i^*} \frac{1}{\tilde{\Delta}_i^2} \log\left(\frac{k}{\delta \tilde{\Delta}_i}\right)\right)$ and $\Omega\left(\sum_{i \in [k] \setminus \{i^*\}} \frac{1}{\max\left(\tilde{\Delta}_i, \alpha\right)^2} \log \frac{1}{\delta}\right)$ respectively (Theorems 3 and 18 therein), where $\tilde{\Delta}_i$ is the gap of arm $i$ in the contaminated case, $[k]$ is the set of of arms, $\delta$ and $\alpha$ are the confidence and accuracy parameters, and

$i^*$ is the optimal arm. The methods introduced for the contaminated best arm identification problem in Altschuler et al. (2019) operate on the principle of elimination, where designs that are not the best arm are progressively removed until one arm remains. However, this termination criterion, and hence the methods of Altschuler et al. (2019), cannot be trivially extended to the multi-objective setting, as the number of Pareto optimal arms is unknown. To address these challenges, we establish the median statistic as a robust measure in the multi-objective case and propose a method to solve PSI in the adversarial setting. The detailed comparison of our work with the prior work from the literature is provided in Table 1. Our contributions are summarized as follows:

- We propose a robust algorithm that returns an $(\alpha, \delta)$-PAC Pareto set of arms under adversarial contamination, including oblivious, prescient and malicious adversarial attacks (see Section 3.1 for precise definitions).

- We provide a tight sample complexity bound for our algorithm depending on the contamination probability. In particular, when the reward distributions are subgaussian, our sample complexity bound has the same dependence on $\alpha$ as Algorithm 1 of Auer et al. (2016) that works in the contamination-free setting, scaling as $\mathcal{O}\left(\frac{K}{\alpha^2} \log\left(\frac{MK}{\delta\alpha}\right)\right)$.

- We conduct extensive experiments on real world data that verify the robustness of our algorithm to adversarial corruptions.

### 1.2 Organization

In Section 2, we introduce the necessary notation. In Section 3, we formulate the adversarial MO-MAB problem. In Section 4, we describe our median-based Pareto elimination algorithm. In Section 5, we prove that the proposed algorithm satisfies the accuracy and coverage requirements defined in Section 3.4 and prove a sample complexity bound. In Section 6, we give the experimental results. Conclusions of the research and future directions are highlighted in Section 7.

## 2 Notation

We denote the Bernoulli distribution with parameter $\rho \in [0, 1]$ by $\text{Ber}(\rho)$. We denote the set of positive integers by $\mathbb{N}_+$ and the set $\{1, ..., n\}$ by $[n]$ for $n \in \mathbb{N}_+$. We use the short hand notation $[a \pm b]$ to denote the interval $[a - b, a + b]$. We use the abbreviation *w.h.p.* to denote *with high probability* and *cdf* to denote *cumulative distribution function*.

Let $F$ represent a cdf and $X$ be a random variable such that $X \sim F$. We denote the right and left quantile functions of $X$ by $Q_{R,F}(p) := \inf\{x \in \mathbb{R} : F(x) > p\}$ and $Q_{L,F}(p) := \inf\{x \in \mathbb{R} : F(x) \geq p\}$ for $p \in [0, 1]$ respectively. The following notations are borrowed from Altschuler et al. (2019). The set of medians is denoted by $m_1(F) := [Q_{L,F}(\frac{1}{2}), Q_{R,F}(\frac{1}{2})]$. We also use the shorthand $m_1(X)$ to denote $m_1(F)$. In the case where median is unique, we use $m_1(F)$ to denote the median instead of the singleton set containing this value. Note that $m_1(F)$ can be considered to be the robust analogue to mean. We denote the empirical median of a sequence of samples $x_1, \ldots, x_n \in \mathbb{R}$ as $\hat{m}_1(x_1, \ldots, x_n)$. If $n$ is odd, this corresponds to the middle value in the sequence. If $n$ is even, it corresponds to the average of two middle values. Given that $F$ has a unique median, we define the median absolute deviation of $F$ as $m_2(F) := m_1(|X - m_1(F)|)$.

Suppose $x$ is an $M$-dimensional vector. We denote the $i$th element of $x$ by $x^i$. Consider another $M$-dimensional vector $y$. We use the notation $x \preceq y$ to denote that vector $x$ is weakly dominated by vector $y$, or equivalently, for all $i \in [M] : x^i \leq y^i$. Also we use the notation $x \npreceq y$ to denote that $x$ is not weakly dominated by $y$, or equivalently, there exists $i \in [M] : x^i > y^i$. We say that $x$ is dominated by $y$, denoted by $x \prec y$, if $x \preceq y$ and there exists $i \in [M] : x^i < y^i$.

Suppose $a$ is a scalar and $x$ is a vector. We use the notation $x + a$ and $x - a$ to denote the summation of each element of $x$ with $a$ and the subtraction of each element of $x$ by $a$ respectively. We also define the ordering relations between scalars and vectors similar to the ones defined between the vectors above: $x \preceq a$ denotes that for all $i$, $x^i \leq a$; $a \preceq x$ denotes that for all $i$, $a \leq x^i$; $a \npreceq x$ denotes that there exists $i$ such that $a > x^i$ and $x \npreceq a$ denotes that there exists $i$ such that $x^i > a$.

# 3 Problem Formulation

We consider a multi-objective pure exploration problem with $M$ objectives indexed by $d \in [M]$ and $K$ arms indexed by $i \in [K]$. The learner sequentially samples arms over rounds $n \in \mathbb{N}_+$. When selected in round $n$, arm $i$ generates a random reward (outcome) vector $Y_{i,n} := (Y_{i,n}^d)_{d \in [M]}$, where $Y_{i,n}^d$ represents the reward in objective $d$. Arm reward distributions do not depend on $n$. The cdf of $Y_{i,n}^d$ is denoted by $F_i^d$. The random variables $Y_{i,n}^a$ and $Y_{j,n}^b$ can be correlated for all $i, j \in [K]$ and $a, b \in [M]$. When an arm is selected, its random reward vector is not directly observed by the learner. Instead, the learner observes the contaminated random reward vector denoted by $\tilde{Y}_{i,n} := (\tilde{Y}_{i,n}^d)_{d \in [M]}$, where $\tilde{Y}_{i,n}^d$ represents the contaminated reward in objective $d$.

Next, we describe the contamination model. The contamination probability is fixed across all arms and objectives and is denoted by $\epsilon \in (0, \frac{1}{2})$. Bernoulli random variable corresponding to arm $i$ and objective $d$ that determines whether a contamination occurs at round $n$ is denoted by $B_{i,n}^d \sim \text{Ber}(\epsilon)$. If a contamination occurs at arm $i$ and objective $d$ in round $n$, then the observed reward is sampled from the contamination distribution instead of the true reward distribution. Formally,

$$\tilde{Y}_{i,n}^d = \begin{cases} Y_{i,n}^d & \text{if } B_{i,n}^d = 0 \\ Z_{i,n}^d & \text{if } B_{i,n}^d = 1 \end{cases}$$

where $Z_{i,n}^d$ represents the contaminated reward. Equivalently, $\tilde{Y}_{i,n}^d = (1 - B_{i,n}^d)Y_{i,n}^d + B_{i,n}^d Z_{i,n}^d$. The cdf of $Z_{i,n}^d$ is denoted by $G_{i,n}^d$. We allow the contamination distributions depend on round $n$. The cdf of $\tilde{Y}_{i,n}^d$ is denoted by $\tilde{F}_{i,n}^d$.

We define *median of interest* $m_i^d$, corresponding to cdf $F_i^d$, as the mean of right and left $(1/2)$-quantiles of $F_i^d$, i.e.,

$$m_i^d := \frac{Q_{R,F_i^d}(\frac{1}{2}) + Q_{L,F_i^d}(\frac{1}{2})}{2} .$$

Note that if $F_i^d$ has a unique median, $m_i^d$ is equivalent to this median. We also define *median of interest vector* of arm $i$, which we denote by $m_i$, as the $M$-dimensional vector whose elements are the medians of interests that are associated with arm $i$. Next, we define the Pareto optimal set of arms according to median of interest.

**Definition 1.** $P^* := \{i \in [K] | \nexists j \in [K] : m_i \prec m_j\}$.

The learner's goal is to (approximately) identify $P^*$ by sampling as few arms as possible. A PSI algorithm stops after conducting a series of sequential evaluations of arms in $[K]$ with the aim of returning a predicted Pareto set $P$ that approximates $P^*$ up to a given level of accuracy (formally defined in Section 3.3).

## 3.1 Adversarial Contamination Models

We consider three contamination models, which we give in the order from the weakest to the strongest below in terms of the adversarial power. Our contamination models are extensions of the contamination models in Altschuler et al. (2019) to MO-MAB. Specifically, we extend their contamination models to hold for any dimension $d \in [M]$.

*Oblivious adversary:* Chooses all the contamination distributions a priori without the knowledge of the arm rewards or the rounds in which the samples are corrupted. Formally, for any given $i \in [K]$ and all $d \in [M]$, $\{(Y_{i,n}^d, Z_{i,n}^d, B_{i,n}^d)\}_{n \geq 1}$ triples are independent. Furthermore, for any given $n$, $Y_{i,n}^d$ and $B_{i,n}^d$ are independent for all $i$ and $d$. Therefore, $\tilde{F}_{i,n}^d$ is equivalent to $(1 - \epsilon)F_i^d + \epsilon G_{i,n}^d$ in this model. A motivating example for oblivious adversary is sensing errors in sensor networks. An arm corresponds to a sensor, and once activated the sensor collects $M$ measurements. Oblivious adversary models randomly occurring measurement errors due to the environmental effects or sensor defects.

*Prescient adversary:* Can choose contamination distributions based on all the past and future true arm rewards and the outcome of Bernoulli random variable that determines if a contamination occurs. Formally, for any given $i \in [K]$ and $d \in [M]$, the pairs $\{(Y_{i,n}^d, B_{i,n}^d)\}_{n \geq 1}$ are independent. Furthermore, for

any given $n$, $Y_{i,n}^d$ and $B_{i,n}^d$ are independent for all $i$ and $d$ and $Z_{i,n}^d$ may depend on all the realizations $\{Y_{j,s}^d, B_{j,s}^d, Z_{j,s}^d\}_{j \in [K], d \in [M], s \geq 1}$. This can model randomly occurring sensing errors where the observed (corrupted) value depends on the true rewards. The prescient adversary model can also be a good fit for review bombing. For instance, a protesting user can give lowest scores to Pareto optimal arms.

*Malicious adversary:* In the malicious adversary, not only can the adversary choose the contamination distributions based on both past and future true arm rewards (as in the prescient model), but they also have the additional ability to manipulate the occurrence of contaminations based on the true rewards.

Formally, in the case of prescient adversary, for any arm $i \in [K]$ and objective $d \in [M]$, the pairs $\{(Y_{i,n}^d, B_{i,n}^d)\}_{n \geq 1}$ remain independent over time. However, unlike the prescient adversary, the malicious adversary can couple the contamination indicator $B_{i,n}^d$ (which determines whether a reward is contaminated) with the true reward $Y_{i,n}^d$. This means that the adversary can condition both the choice of whether to contaminate and the contaminated value $Z_{i,n}^d$ on all previous and future observed outcomes.

For example, in a cybersecurity setting, an attacker can perform a man-in-the-middle attack, where they intercept and modify communications between two parties. By selectively corrupting rewards from certain arms, the attacker can manipulate the algorithm's choices.

### 3.2 Unavoidable Bias and Median Concentration

Because our adversarial contamination models allow for an arbitrary contamination distribution, mean statistics cannot be predicted from the contaminated samples. Furthermore, median statistic is subject to an unavoidable bias which makes the median identifiable only up to a certain interval. Below we will review results from Altschuler et al. (2019), which quantifies the amount of unavoidable bias and the concentration of sample median. The results presented in this subsection are related to the concentration of the median in a single objective. Later on, we will utilize them for our multi-objective PSI identification and sample complexity analysis.

**Definition 2.** *(Altschuler et al., 2019, Definition 5). For any $\bar{t} \in (0, \frac{1}{2})$ and positive, non-decreasing function $R$ defined on domain $[0, \bar{t}]$, define $C_{R,\bar{t}}$ to be the family of all distributions $F$ that satisfy the following:*

$$R(t) \geq \max \left\{ Q_{R,F}\left(\frac{1}{2} + t\right) - m, m - Q_{L,F}\left(\frac{1}{2} - t\right) \right\} \text{ for all } t \in [0, \bar{t}] \text{ and } m \in m_1(F).$$

$R$ bounds the maximum quantile deviation that can occur from the median. It will play a key role in our sample complexity analysis. We will choose a common $R$ for all $\{F_i^d\}_{i,d}$ in order to facilitate our analysis.

Below, we state results on concentration of the empirical median. These will be used in our sample complexity analysis.

**Lemma 1.** *(Upper bound on empirical median deviation for prescient and oblivious adversaries) (Altschuler et al., 2019, Lemma 7). Let $\bar{t} \in (0, \frac{1}{2})$, $\epsilon \in (0, \frac{2\bar{t}}{1+2\bar{t}})$, $\delta \in (0, 1)$ and $F \in C_{R,\bar{t}}$, where $R$ is a non-decreasing function defined on domain $[0, \bar{t}]$. Let $Y_i \sim F$ and $B_i \sim Ber(\epsilon)$ all be independently drawn for $i \in [n]$. Let $\{Z_i\}_{i \in [n]}$ be arbitrary random variables possibly depending on $\{Y_i, B_i\}_{i \in [n]}$, and $\tilde{Y}_i = (1 - B_i)Y_i + B_i Z_i$. Then for $n \geq 2(\bar{t} - \frac{\epsilon}{2(1-\epsilon)})^{-2} \log(\frac{2}{\delta})$:*

$$\mathbb{P}\left( \sup_{m \in m_1(F)} |\hat{m}(\tilde{Y}_1, ..., \tilde{Y}_n) - m| \leq R\left(\frac{\epsilon}{2(1-\epsilon)} + \sqrt{\frac{2\log(2/\delta)}{n}}\right)\right) \geq 1 - \delta .$$

**Lemma 2.** *(Upper bound on empirical median deviation for malicious adversary) (Altschuler et al., 2019, Lemma 8). Let $\bar{t} \in (0, \frac{1}{2})$, $\epsilon \in (0, \bar{t})$, $\delta \in (0, 1)$ and $F \in C_{R,\bar{t}}$, where $R$ is a non-decreasing function defined on domain $[0, \bar{t}]$. Let $(Y_i, B_i)$ pairs be independently drawn for $i \in [n]$ with marginals $Y_i \sim F$ and $B_i \sim Ber(\epsilon)$. Let $\{Z_i\}_{i \in [n]}$ be arbitrary random variables possibly depending on $\{Y_i, B_i\}_{i \in [n]}$, and $\tilde{Y}_i = (1 - B_i)Y_i + B_i Z_i$. Then for $n \geq 2(\bar{t} - \epsilon)^{-2} \log(\frac{3}{\delta})$:*

$$\mathbb{P}\left( \sup_{m \in m_1(F)} |\hat{m}(\tilde{Y}_1, ..., \tilde{Y}_n) - m| \leq R\left(\epsilon + \sqrt{\frac{2\log(3/\delta)}{n}}\right)\right) \geq 1 - \delta .$$

In the expressions above, we observe that in the limiting case as $n \to \infty$, the difference between sample median and median is upper bounded by $D := R(\frac{\epsilon}{2(1-\epsilon)})$ for oblivious and prescient adversaries and $D := R(\epsilon)$ for malicious adversary. An intriguing question is whether these bounds can be improved. The answer is negative, as (Altschuler et al., 2019, Corollary 6 & Lemma 9) shows that there exists reward and contamination distributions for which $D$ is unavoidable. Therefore, as in Altschuler et al. (2019), we call $D$ the unavoidable bias term.

Note that the above lemmas can be used for bounding the deviation of empirical median from the median of interest since median of interest is also a median of the given distribution. In the rest of the paper, we will simply refer to *median of interest* as the *median* and the *median of interest vector* as the *median vector*.

The results presented above are very general. In the examples below, we show how $R$ can be defined for specific families of distributions. Below, we provide a suitable $R$ for subgaussian distributions, which are commonly used in bandit problems.

**Example 1.** *All $\sigma$-subgaussian distributions are members of the family $C_{R,\bar{t}}$, where $\bar{t} \in (0, 1/2)$ and*

$$R(t) = \sigma\sqrt{2}\left(\sqrt{\log\left(\frac{1}{1/2-t}\right)} + \sqrt{\log(2)}\right). \tag{1}$$

Derivation of equation 1 can be found in Appendix A.1.

Another interesting case, which allows sharper bounds is the family of distributions whose cdfs increase linearly around the median (not too flat around the median).

**Definition 3.** *(Altschuler et al., 2019, Definition 10) Given $\bar{t} \in (0, 1/2)$ and $B > 0$, define $\mathcal{F}_{B,\bar{t}}$ as the family of distributions $F$ with a unique median such that for all $x_1, x_2 \in [Q_{L,F}(1/2 - \bar{t}), Q_{R,F}(1/2 + \bar{t})]$*

$$|F(x_1) - F(x_2)| \geq \frac{1}{Bm_2(F)}|x_1 - x_2|.$$

As noted in Altschuler et al. (2019), (i) any univariate Gaussian distribution is in $\mathcal{F}_{B,\bar{t}}$ for any $\bar{t} \in (0, 1/2)$ and $B \geq q_{3/4}/\phi(q_{1/2+\bar{t}})$, where $\phi$ is the standard Gaussian density and $q_\alpha$ is its $\alpha$ quantile, (ii) any uniform distribution defined on an interval is in $\mathcal{F}_{B,\bar{t}}$ for any $\bar{t} \in (0, 1/2)$ and $B \geq 4$. Moreover, all distributions $F$ in $C_{R,\bar{t}}$ given in Definition 3 satisfy Definition 2 with $R(t) = Bm_2(F)t$.

### 3.3 Multi-objective Suboptimality Gap

The number of samples required to distinguish an arm $i \notin P^*$ from an arm $j \in P^*$ depends on distance between arms $i$ and $j$. We quantify this distance by the notion of suboptimality gap.

**Definition 4.** *We define $\Delta_{i,j} := \max\{0, \min_d(m_j^d - m_i^d)\}$ as the suboptimality gap of an arm $i$ with respect to arm $j$ and $\Delta_i := \max_{j \in P^*} \Delta_{i,j}$ as the suboptimality gap of arm $i$.*

$\Delta_i$ measures how much arm $i$ is dominated by the Pareto set. Given a positive real number $\alpha$, we call an arm $i$ $\alpha$-optimal if $\Delta_i \leq \alpha$, and $\alpha$-suboptimal if $\Delta_i > \alpha$. All Pareto optimal arms are $\alpha$-optimal for any positive real number $\alpha$ since $\Delta_j = 0$ for a Pareto optimal arm $j$.

Due to the unavoidable bias, in general, it is not possible to detect Pareto optimal arms with more than $2D$ accuracy, as proven in the following remark, whose details can be found in Appendix A.2.

**Lemma 3.** *Suppose that an adversary can alter a reward sample as much as $D$ so that either $\lim_{n \to \infty} \hat{m}(\tilde{Y}_{i,1}^d, \cdots, \tilde{Y}_{i,n}^d) = m + D$ or $\lim_{n \to \infty} \hat{m}(\tilde{Y}_{i,1}^d, \cdots, \tilde{Y}_{i,n}^d) = m - D$ holds for $i \in [K]$. Then, given any $\zeta > 0$, there are bandit environments in which it is impossible to distinguish $(2D - \zeta)$-optimal arms from the Pareto optimal arms.*

### 3.4 Pareto Accuracy

In the following, we define the class of algorithms that is of interest to us in the adversarial MO-MAB setting.

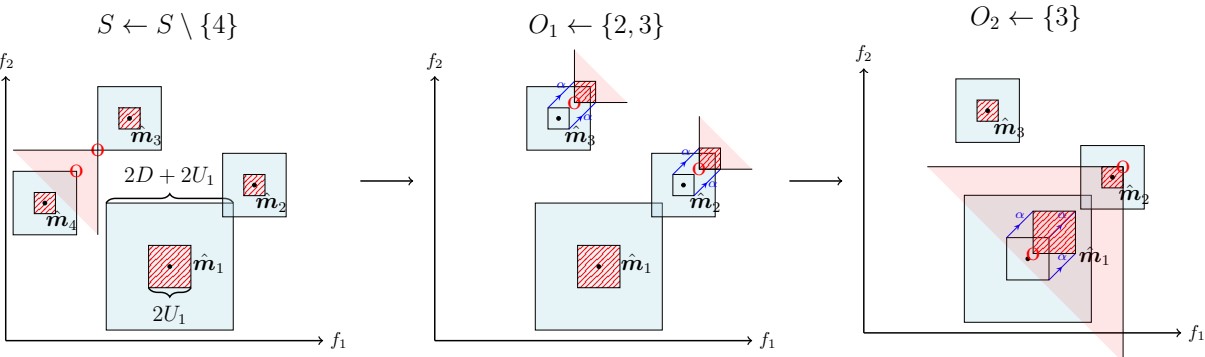

Figure 1: Visualization of R-PSI algorithm in two dimensional objective space. The red circles mark points used in comparisons. The blue squares are the confidence regions of arms given in Lemma 4, whereas shaded squares are the smaller regions used in identification phase. The left figure visualizes the elimination phase, where arm 3 eliminates arm 4. The middle and right figures visualize the Pareto identification phase of the algorithm. Both arm 2 and arm 3 are added to $O_1$, as shown in the middle figure. However, arm 2 is suspected to be useful in elimination of the arm 1 in the future rounds, thus it is not added to $O_2$ as depicted in the right figure. As a result, only arm 3 is added to the estimated Pareto set $P$.

**Definition 5.** *(Pareto accurate algorithm) Suppose that the reward distributions belong to $C_{R,\bar{t}}$ given in Definition 2 for some $\bar{t} \in (0, \frac{1}{2})$. Then, given the accuracy parameter $\alpha \geq 0$, the confidence probability $0 < \delta < 1$ and the adversarial attack probability $0 \leq \epsilon < \frac{2\bar{t}}{1+2\bar{t}}$ for oblivious and prescient adversaries, $0 \leq \epsilon < \bar{t}$ for malicious adversaries, we call an algorithm Pareto accurate in the adversarial MO-MAB setting, if the set of arms $P$ that the algorithm returns at termination satisfies the following conditions:*

1. *Accuracy: All the returned arms are $(2D + \alpha)$-optimal:*

$$\forall i \in P, \ \Delta_i \leq 2D + \alpha .$$

2. *Coverage: If a Pareto optimal arm $j$ is not in $P$, then, there exists at least one arm in $P$ that $(2D)$-covers arm $j$:*

$$\forall j \in P^*, \ \exists i \in P : m_j - m_i \preceq 2D .$$

The Pareto accuracy can be considered as the generalization of the *probably approximately correct* (PAC) learning concept from the single objective MAB setting to the adversarial MO-MAB setting. However, in the adversarial MO-MAB setting, since the Pareto optimal arms cannot be distinguished from other $2D$-optimal arms as shown in Lemma 3, it is not possible to approximate the optimal solution arbitrarily well. This reflects itself in the accuracy and the coverage conditions defined above. We also note that, from the algorithms that are in the class defined above, the ones that have smaller sample complexity would be favorable since in many practical settings, taking samples induce some type of cost, e.g., monetary and time, that we would want to minimize.

## 4 A Robust Learning Algorithm

We propose a *Pareto accurate* algorithm called *Robust Pareto Set Identification* (R-PSI) whose pseudocode is given in Algorithm 1. The algorithm sequentially operates over sampling phases indexed by $t \geq 0$. It keeps two sets of arms: the *undecided set $S$* and the *predicted Pareto set $P$*, where $S \cup P = [K]$ for all $t$. A visualization of the R-PSI algorithm is provided in Figure 1.

At the beginning ($t = 0$), all the arms are assigned to the *undecided set $S$*. Then each arm is sampled $n_0$ times, which is defined as:

$$n_0 := \left\lceil 2\beta_{\bar{t},\epsilon} \log\left(\frac{\pi^2 MK}{6\tilde{\delta}}\right)\right\rceil \tag{2}$$

where

$$\beta_{\bar{t},\epsilon} := \left(\bar{t} - \frac{\epsilon}{2(1-\epsilon)}\right)^{-2}, \ \tilde{\delta} := \frac{\delta}{2} \tag{3}$$

for prescient and oblivious adversaries, and

$$\beta_{\bar{t},\epsilon} := (\bar{t} - \epsilon)^{-2}, \ \tilde{\delta} := \frac{\delta}{3} \tag{4}$$

for malicious adversary.

For sampling phase $t \geq 1$, let $\tau_i$ represent the number of sampling phases so far in which arm $i$ is sampled. At each sampling phase $t \geq 1$, the algorithm selects the arm with the largest statistical bias which we denote by $i^*$. Computation of $i^*$ is closely linked with Lemmas 1 and 2. In particular, given $R$, the statistical bias of any arm after $\tau$ sampling rounds represents the uncertainty in the median estimate attributed to sample size, and is computed as

$$U_\tau = R\left(h_\epsilon + 1/\sqrt{\beta_{\bar{t},\epsilon}\tau}\right) - R(h_\epsilon) \ ,$$

where $h_\epsilon := \frac{\epsilon}{2(1-\epsilon)}$ for oblivious and prescient adversary and $h_\epsilon := \epsilon$ for malicious adversary. Thus, we set

$$U_i = U_{\tau_i} \ , \tag{5}$$

and $i^* = \arg\max_{k \in [K]} U_{k,\tau_k}$.

**Remark 1.** *The difference between the sampling round numbers of any two arms in $S$ cannot be larger than 1 since the algorithm selects the arm with the largest statistical bias at each sampling phase.*

$$\forall i, j \in S, \ |\tau_i - \tau_j| \leq 1 \ .$$

After arm $i^*$ is selected, it is successively sampled $n_{\tau_{i^*}}$ times where

$$n_{\tau_{i^*}} := 1 + \left\lceil 4\tau_{i^*}\beta_{\bar{t},\epsilon}\log\left(\frac{\tau_{i^*}}{\tau_{i^*} - 1}\right) + 2\beta_{\bar{t},\epsilon}\log\left(\frac{(\tau_{i^*} - 1)^2 MK\pi^2}{6\tilde{\delta}}\right)\right\rceil \ .$$

As will be proven in Lemma 4, $n_{\tau_{i^*}}$ is chosen in such a way that the sample complexity requirements of Lemma 1 and Lemma 2 are met and the resulting empirical median deviation bound of Lemma 1 and Lemma 2 depends on $\tau_{i^*}$ through the expression $R\left(h_\epsilon + \frac{1}{\sqrt{\beta_{\bar{t},\epsilon}\tau_{i^*}}}\right)$.

After each sampling phase, algorithm enters the elimination step where the arms that are guaranteed to be $(2D + \alpha)$-suboptimal are eliminated. As shown in Lemma 5, none of the Pareto optimal arms can be eliminated at this step which is crucial for our theoretical analysis to hold since the *additive property of suboptimality* is not satisfied in the adversarial setting as shown in Remark 4 of Altschuler et al. (2019).

After the elimination step, algorithm enters the identification step where the arms that are guaranteed to satisfy the accuracy requirements are collected in $O_1$. Among these arms in $O_1$, the ones that can potentially eliminate an arm in $S$ at future rounds are dropped back to $S$. If these arms were to remain in $O_1$, the $(2D + \alpha)$-suboptimal arms to be eliminated by these arms will never be eliminated and might be returned in $P$ by the algorithm. This prevents arms that are $(2D + \alpha)$-suboptimal to potentially end up in $P$. The rest of the arms in $O_1$ are collected in $O_2$.

Note that we have an if statement in the identification step that checks whether $U_k > \alpha/4$ and when $U_k \leq \alpha/4$, the algorithm terminates by moving all the arms in $O_1$ to $P$. This is to guarantee the termination

---

**Algorithm 1** R-PSI

1: **Input:** $\alpha$, $\delta$, $\epsilon$, $\bar{t}$, $R(\cdot)$
2: **Initialize:** $S = [K]$, $P = \emptyset$, $\tau_i = 0 \;\; \forall i \in [K]$, $t = 0$.
3: $\tau_i \leftarrow \tau_i + 1 \;\forall i \in [K]$
4: Sample each arm $n_0$ times as in (2)
5: Update $U_i$ according to (5) $\;\forall i \in S$
6: Update $\hat{m}_i \;\;\forall i \in S$

7: **while** $S \neq \emptyset$ **do**
8:    **if** $t > 0$ **then**
9:       **Sampling:**
10:       Choose arm $i^* = \arg\max_{k \in [K]} U_{k,\tau_k}$
11:       $\tau_{i^*} \leftarrow \tau_{i^*} + 1$
12:       Sample $i^*$ successively $n_{\tau_{i^*}}$ times.
13:       Update $U_{i^*}$ according to (5)
14:       Update $\hat{m}_i^*$
15:    **end if**

16:    **Elimination:**
17:    $S \leftarrow S \setminus \{i \in S | \exists j \in S \setminus \{i\} : \hat{m}_i + D + U_i \prec$

18:          $\hat{m}_j - D - U_j\}$
19:    **Identification:**
20:    $O_1 \leftarrow \{i \in S | \nexists j \in S \setminus \{i\} : \hat{m}_i - U_i + \alpha \preceq$
21:       $\hat{m}_j + U_j\}$
22:    **if** $\exists k \in S : U_k > \alpha/4$ **then**
23:       $O_2 \leftarrow \{i \in O_1 | \nexists j \in S \setminus \{i\} : \hat{m}_j - U_j + \alpha \preceq$
24:          $\hat{m}_i + U_i\}$
25:       $S \leftarrow S \setminus O_2$
26:       $P \leftarrow P \cup O_2$
27:    **else**:
28:       $P \leftarrow P \cup O_1$
29:       **return** $P$
30:    **end if**
31:    $t \leftarrow t + 1$
32: **end while**
33: **return** $P$

---

of the algorithm as there might be some arms left in $S$ with suboptimality gaps between $(4D + \alpha)$ and $(2D + \alpha)$ that might cause algorithm to stuck in an infinite loop in the absence of this step. By Lemma 9, arms that are $(4D + \alpha)$-suboptimal are eliminated by the R-PSI algorithm in the earlier rounds. To align with this guarantee, we define $\bar{\Delta}_i := \Delta_i - 4D$ as the adjusted suboptimality gap of arm $i$, indicating that arms with $\bar{\Delta}_i > \alpha$ are eliminated earlier due to their significant suboptimality.

# 5 Accuracy and Sample Complexity Analysis

In this section, we provide accuracy and sample complexity analysis for R-PSI.

## 5.1 Good Event

We start by showing that the *good event* in which the sample median concentrates sufficiently around the true median occurs with high probability. The rest of our analysis is based on this *good event*. The details of this result can be found in Appendix A.3.

**Lemma 4.** *Define $E$ as the event in which for all $i \in [K]$, $d \in [M]$ and $\tau_i \geq 1$, the following is satisfied:*

$$m_i^d + D + U_{\tau_i} \geq \hat{m}_i^d \geq m_i^d - D - U_{\tau_i} \;,$$

*or equivalently, $|\hat{m}_i^d - m_i^d| \leq D + U_{\tau_i}$. Then:*

$$\mathbb{P}(E) \geq 1 - \delta \;.$$

## 5.2 Main Results

In this section, we present the theoretical analysis of R-PSI for the Pareto set identification problem in MO-MAB with contaminated reward observations. A key algorithmic innovation of R-PSI lies in its intricate set operations, which allow it to overcome termination condition issues that hinder previous methods. Through rigorous analysis tailored for the set operations of R-PSI, we prove that, using these set operations, R-PSI returns an $(\alpha, \delta)$-PAC Pareto set, providing strong accuracy guarantees (see Lemmata 5–8). We then establish upper bounds on the sample complexity of R-PSI (see Theorem 1 and Corollaries 1 and 2). These

sample complexity upper bounds rely on the novel termination condition of R-PSI (line 22 of Algorithm 1), which introduces a new approach to determining when the algorithm should stop. This termination criterion leverages a statistical bias term, $U_k$, to ensure that sufficient information has been gathered about the arms before halting. In the proof of Lemmata 5–8, we demonstrate that this termination condition is sufficient to guarantee that R-PSI is a Pareto accurate method. For the sample complexity upper bounds of subgaussian reward distributions (Corollary 1), we use the newly introduced $R(\cdot)$ established in Example 1 in Section 3.2. We also discuss the tightness of the upper bounds in Remark 2.

Next, we state our first main result, which provides an accuracy guarantee for R-PSI and establishes a high-probability upper bound on its sample complexity.

**Theorem 1.** *Assume that the reward distributions belong to $C_{R,\bar{t}}$ given in Definition 2 and the event $E$ defined in Lemma 4 holds. Then, given any $\alpha \in \mathbb{R}_+$, and $\epsilon \leq \frac{2\bar{t}}{1+2\bar{t}}$ for the oblivious and prescient adversary ($\epsilon \leq \bar{t}$ in the case of the malicious adversary), R-PSI is Pareto accurate with sample complexity $N$ bounded by:*

$$
N \leq \sum_{i:\, \Delta_i > 4D + \alpha} 2\tau_{(\bar{\Delta}_i)} \left( \beta_{\bar{t},\epsilon} \log\left( \frac{\tau_{(\bar{\Delta}_i)}^2 MK\pi^2}{6\tilde{\delta}} \right) + 1 \right) + \sum_{i:\, \Delta_i \leq 4D + \alpha} 2\tau_{(\alpha)} \left( \beta_{\bar{t},\epsilon} \log\left( \frac{\tau_{(\alpha)}^2 MK\pi^2}{6\tilde{\delta}} \right) + 1 \right) \quad (6)
$$

$$
\leq K\tau_{(\alpha)} \left( 2\beta_{\bar{t},\epsilon} \log\left( \frac{\tau_{(\alpha)}^2 MK\pi^2}{6\tilde{\delta}} \right) + 2 \right) , \quad (7)
$$

*where $\bar{\Delta}_i := \Delta_i - 4D$, and $\tau_{(a)} := \inf\{\tau : U_\tau \leq a/5\}$ for $a \in \mathbb{R}_+$.*

The above theorem gives the most general expression for the sample complexity without making any further assumptions on the reward distributions. If a suitable $R$ can be determined, for the given reward distributions, then it is possible to derive an explicit gap-dependent bound on the sample complexity. Below, we provide such a result for subgaussian reward distributions.

**Corollary 1.** *Suppose that the reward distributions are subgaussian with parameter $\sigma$. Then, the asymptotic sample complexity (as $\alpha \to 0$) is given by:*

$$
\mathcal{O}\left( \frac{\sigma^2 K \beta_{\bar{t},\epsilon}}{\alpha^2 \log\left( \frac{1}{1/2 - h_\epsilon} \right)} \log\left( \frac{\sigma^2 MK}{\alpha^2 \log\left( \frac{1}{1/2 - h_\epsilon} \right) \tilde{\delta}} \right) \right) . \quad (8)
$$

The sample complexity bound derived for subgaussian distributions in equation 8 has a worst case asymptotic matching to the bounds derived in Theorem 4 of Auer et al. (2016) and Theorem 3 of Altschuler et al. (2019) in terms of $\alpha$-dependence. Also, this bound nearly matches, in the worst case, the adversary-free lower bound in Theorem 17 of Auer et al. (2016). We also note that unlike these studies, our bound contains an $\epsilon$-dependent factor that comes from the adversarial attack. Details about Theorem 1 and Corollary 1 can be found in Appendices A.4 and A.5, respectively.

Based on Definitions 2 and 3, if $F \in \mathcal{F}_{B,\bar{t}}$, then $F \in \mathcal{C}_{R,\bar{t}}$, where $R(t) = Bm_2(F)t$ (Altschuler et al., 2019). In the next Corollary, we provide sample complexity analysis for reward distributions in $F \in \mathcal{F}_{B,\bar{t}}$.

**Corollary 2.** *Suppose that the reward distributions are from the family $F \in \mathcal{F}_{B,\bar{t}}$. Let $\bar{m}_2 \geq max_{i \in [K]} m_2(F)$ for $F \in \mathcal{F}_{B,\bar{t}}$ be a known upper bound. When $R(t) = B\bar{m}_2 t$ is used, R-PSI is Pareto accurate with sample complexity $N$ bounded by*

$$
N \leq \sum_{i}^{K} 2\left( \beta_{\bar{t},\epsilon} + \frac{25\bar{m}_2^2 B^2}{(\Delta_i^\alpha)^2} \right) \left( \log\left( \frac{\left( 1 + \frac{25\bar{m}_2^2 B^2}{\beta_{\bar{t},\epsilon}(\Delta_i^\alpha)^2} \right)^2 MK\pi^2}{6\tilde{\delta}} \right) + 1 \right) ,
$$

where $\Delta_i^\alpha := \max(\alpha, \bar{\Delta}_i)$. Notice that as $h_\epsilon$ approaches $\bar{t}$, the bound increases towards infinity. This is expected as $\bar{t}$ is the threshold for $h_\epsilon$. When $h_\epsilon > \bar{t}$, by Definition 2, $R(t)$ no longer bounds the maximum

possible deviation from medians ((Altschuler et al., 2019), Lemma 1). Hence distinguishing between the contaminated median and the true median becomes uncontrollable. This is reflected by the sample complexity requirements of Lemmata 1 and 2. When $B$, $\bar{m}_2$, $\bar{t}$ are taken as non-negative constants and $\epsilon = 0$, the sample complexity bound takes the form $\mathcal{O}\left(\frac{K}{\alpha^2} \log\left(\frac{MK}{\bar{\delta}\alpha}\right)\right)$ which recovers the worst case bound from Theorem 4 of Auer et al. (2016) for adversary free Pareto set identification in MO-MAB setting. Details about Corollary 2 can be found in Appendix A.6.

**Remark 2.** *To analyze the tightness of the upper bounds, we consider a specific problem instance where $M = 1$. In this case, $P^* = \arg\max_i m_i$ and $\Delta_i = \max_j m_j - m_i$. Thus, the accuracy condition for Pareto accurate algorithms given in Definition 5 reduces to $\forall i \in P, m_i \geq \max_j m_j - 2D - \alpha$. The coverage condition reduces to $\forall j \in P^*, \exists i \in P : m_j - m_i \leq 2D$. Success conditions considered in Altschuler et al. (2019) require that any successful algorithm, for any $\alpha \geq 0$, should return a single arm $I$ such that $m_{i^*} - m_I - U_{i^*} - U_I \leq \alpha$, where $i^* = \arg\max_i m_i$. In terms of our notation, this condition is $m_I \geq \max_j m_j - 2D - \alpha$, which is equivalent to our accuracy condition. Thus, we can say that Altschuler et al. (2019) studies a success condition that is weaker than ours. Any lower bound for the success condition of Altschuler et al. (2019) also holds in our case. In particular, the lower bound stated in Theorem 18 of Altschuler et al. (2019) holds, which is given by $\Omega\left(\sum_{i \in [K] \setminus \{i^*\}} \frac{1}{\max\{\Delta_i - 2D, \alpha\}^2} \log(\frac{1}{\delta})\right)$. In terms of dependence on $\alpha$, both Corollary 1 and 2 match this lower bound up to logarithmic terms.*

## 6  Numerical Results

We compare our algorithm with Algorithm 1 from Auer et al. (2016) which considers Pareto set identification problem for the adversary free MO-MAB. We name this algorithm *Auer-A1*. Auer-A1 is a Pareto accurate algorithm in the adversary free setting ($D = 0$) and its ranking of the arms is based on the mean of the distributions instead of the median. We conduct experiments on MovieLens dataset (Harper & Konstan, 2015), SW-LLVM dataset Zuluaga et al. (2013), and data obtained from UVA/PADOVA Type 1 Diabetes Simulator (Man et al., 2014). For the algorithms to be comparable, except MovieLens dataset, we consider Gaussian rewards so that the median and the mean are the same. However, the experiments on MovieLens dataset are an exception, as its reward distribution is categorical, meaning the mean and median values may differ. This discrepancy between the mean and median values might affect the performance of Auer-A1. To mitigate this effect, we choose as the arm set a subset of movies in MovieLens dataset such that the Pareto set in terms of median rewards are the same as the Pareto set in terms of mean rewards. We also compare our algorithm with a modification of Auer-A1 algorithm that uses median values instead of mean values and thus returns a median-based Pareto set. We call this median based method *Auer-A1-M*.

Note that in Auer et al. (2016), the success condition differs from the accuracy and coverage requirements we use to define the Pareto accuracy in the adversarial MO-MAB setting. In particular, their success condition requires the predicted arms to be at least $\alpha$-accurate and all the Pareto optimal arms to be returned in the predicted set. In terms of our accuracy and coverage arguments, this success condition is equivalent to an $\alpha$-accuracy and 0 margin coverage requirement. For a fair comparison, while evaluating Auer-A1 and Auer-A1-M, we relax this success condition to $(2D + \alpha)$ accuracy and $(2D)$ coverage requirement, which are equivalent to the requirements set for R-PSI. In real-world applications, the reward distributions of the arms may not be available. To maintain realism in our experiments, in all experiments, we use the $R$ from Example 1. This is because the subgaussian assumption covers the largest set of cases for which we have derived explicit theoretical results, making it more suitable when dealing with an unknown reward function.

Robust PSI can be viewed as a classification task where the true positives are accurate arms and false negatives are uncovered arms. To align with existing research on classification tasks and to provide a metric that is appropriate for the accuracy and coverage conditions as defined in Definition 5, we define the robust analogue of F1 score and denote it by $r$-F1 score. Our definition is inspired by the $\epsilon$-F1 score introduced by Karagözlü et al. (2024), which is designed for approximate arm identification in the context of vector optimization.

$$r\text{-F1} := \frac{2\left|\Pi_r \cap P\right|}{2\left|\Pi_r \cap P\right| + \left|\Pi_{r \setminus P}\right| + \left|P \setminus \Pi_r\right|} ,$$

where $\Pi_r$ is the set of $2D + \alpha$ optimal arms and $\Pi_{r \setminus P}$ is the set of Pareto optimal arms that is not covered by $P$. Note that $r$-F1 $= 1$ if and only if the algorithm is Pareto accurate (see Definition 5). The experiments also utilize other metrics: SC (average sample complexity), RSR (ratio of successful runs), and RO (ratio of optimal arms). RSR measures the ratio of experiments where the Pareto set is accurately identified under the adversarial success conditions defined in Definition 5. A higher RSR score indicates that the algorithm achieves this task consistently. This metric is crucial in scenarios where robust performance across multiple trials is essential, since a high RSR score corresponds to achieving the success condition in most of the experiments. SC is the average number of samples required by the algorithm to return the estimated Pareto set. A lower SC indicates greater efficiency, as it means the algorithm achieves its result with fewer arm pulls. Comparing SC across different algorithms provides an understanding of the trade-offs between accuracy and efficiency. For example, an algorithm might achieve high accuracy but require significantly more samples. RO is the ratio of the number of Pareto optimal arms returned by the method to the total number of true Pareto optimal arms, i.e., $|P^* \cap P|/|P^*|$. This metric tells us how much of the true Pareto arms were returned. A high RO metric means that the algorithm is returning most of the arms in the true Pareto set. However, this metric alone cannot capture how accurate an algorithm is since an algorithm might achieve a perfect RO score, while returning too many arms that are not true Pareto arms. RO metric might seem redundant since r-F1 metric also considers the recall of the algorithm, but it is still crucial while comparing algorithm performances. For example, in a setting where there are only a few arms in the true Pareto optimal set, an algorithm which returns all $(2D + \alpha)$-optimal arms except the true Pareto arms will achieve a high r-F1 score, but RO will be 0 since none of the true Pareto arms were returned.

## 6.1 Experiments on MovieLens Dataset

Review bombing is an internet phenomenon where a large group of accounts post negative reviews or ratings for a product, service, or content as a form of protest or manipulation. This practice can significantly skew public perception and has prompted many review platforms to develop methods to mitigate its effects. We consider the movie reviews on MovieLens, a movie recommendation service. We select 24 movies and assign each of them to an arm. When an arm is pulled, a randomly selected review of that arm is observed. The objective of the optimization is to find the Pareto arms in terms of the review scores across five age demographics: $0-18, 19-25, 26-35, 36-45, 46+$. (i.e. $M = 5$). To simulate the task of Pareto identification under an event of review bombing, we choose a prescient adversary that replaces the scores of the Pareto optimal arms with the lowest review score (i.e., 1) in contaminated objectives and leaves the non-Pareto arms as is. We choose $\sigma = 0.2$ in equation 1, $\delta = 0.1$, and $\alpha = 0.2$. We use $\bar{t} = 0.49$ and maximum $\epsilon$ value of 0.4. We report the average results over 100 runs in Table 2. The results indicate that Auer-A1's success rate declines as adversaries get stronger, whereas R-PSI consistently makes accurate predictions with fewer samples. Though Auer-A1-M returns an accurate and covering Pareto set, it misses out some of the true Pareto designs indicated by the low RO score.

## 6.2 Experiments on SW-LLVM Dataset

We use SW-LLVM dataset from Zuluaga et al. (2013) which consists of 1023 compiler settings characterized by 11 binary features. The objectives are performance and memory footprint of some software when compiled with these settings. Similar to Auer et al. (2016), to obtain a stochastic-like data for our algorithm, we use the combinations of 4 of the binary features to form 16 arms. By ignoring all the other features we end up with 64 data points for each arm. When an arm is pulled, one of the 64 data points pertaining to that arm is randomly selected and the corresponding objectives are returned as reward. We normalize the data to obtain a similar range for both objectives. We assume that the reward distributions are Gaussian. Note that this assumption is for the purpose of determining the parameters of the algorithm and does not affect the rewards obtained from arm pulls in any way. We take the mean of 64 data points assigned to an arm and use this as the mean (median) of the corresponding Gaussian reward distribution. We select a malicious adversary that is 25% more likely to contaminate Pareto arms compared to non-Pareto arms, while also preserving the marginal distribution $\mathrm{Ber}(\epsilon)$. The contamination has a value of $\pm 10$. The choice for the parameters $\sigma$, $\delta$, and $\bar{t}$ are the same as in the previous setting, while $\alpha$ was chosen to be 0.1. We report the average results over 100 runs in Table 2. It can be seen that contaminations as low as $\epsilon = 0.05$ prevent Auer-A1 from

Table 2: Experiments on real world datasets. From left to right: MovieLens, SW-LLVM dataset and UVA/PADOVA simulator experiment results.

| | $\epsilon$ | **MovieLens** | | | | **SW-LLVM** | | | | **UVA/PADOVA** | | | |
| | | RSR | SC | RO | r-F1 | RSR | SC | RO | r-F1 | RSR | SC | RO | r-F1 |
|---|---|---|---|---|---|---|---|---|---|---|---|---|---|
| | 0.0 | 1.0 | 18674.7 | 1.0 | 1.0 | 1.0 | 1540.9 | 1.0 | 1.0 | 1.0 | 3951.8 | 1.0 | 1.0 |
| | 0.05 | 1.0 | 20425.0 | 1.0 | 1.0 | 1.0 | 54935.1 | 1.0 | 1.0 | 1.0 | 4432.0 | 1.0 | 1.0 |
| R-PSI | 0.1 | 1.0 | 20914.1 | 1.0 | 1.0 | 1.0 | 57822.5 | 1.0 | 1.0 | 1.0 | 5067.2 | 1.0 | 1.0 |
| | 0.2 | 1.0 | 33753.2 | 1.0 | 1.0 | 1.0 | 72969.3 | 1.0 | 1.0 | 1.0 | 7232.8 | 1.0 | 1.0 |
| | 0.3 | 1.0 | 49212.2 | 1.0 | 1.0 | 1.0 | 98714.7 | 1.0 | 1.0 | 1.0 | 12683.8 | 1.0 | 1.0 |
| | 0.4 | 1.0 | 105847.2 | 1.0 | 1.0 | 1.0 | 222906.8 | 0.92 | 1.0 | 1.0 | 39647.0 | 1.0 | 1.0 |
| | 0.0 | 1.0 | 33779.0 | 1.0 | 1.0 | 1.0 | 7175.7 | 1.0 | 1.0 | 1.0 | 642344.1 | 1.0 | 1.0 |
| | 0.05 | 0.0 | 59092.6 | 0.0 | 0.92 | 0.97 | 250885.9 | 0.0 | 0.98 | 1.0 | 900844.7 | 1.0 | 1.0 |
| Auer-A1 | 0.1 | 0.0 | 59428.8 | 0.0 | 0.92 | 0.92 | 174599.9 | 0.0 | 0.97 | 1.0 | 1089962.2 | 0.93 | 1.0 |
| | 0.2 | 0.0 | 58917.8 | 0.0 | 0.92 | 0.78 | 80767.7 | 0.0 | 0.90 | 0.8 | 1450304.8 | 0.71 | 0.99 |
| | 0.3 | 0.0 | 57953.3 | 0.0 | 0.92 | 0.91 | 41946.3 | 0.0 | 0.96 | 0.0 | 1946291.2 | 0.71 | 0.75 |
| | 0.4 | 0.0 | 57765.6 | 0.0 | 0.92 | 0.91 | 37811.9 | 0.0 | 0.97 | 0.0 | 1696773.8 | 0.41 | 0.75 |
| | 0.0 | 1.0 | 33320.7 | 1.0 | 1.0 | 1.0 | 9896.7 | 1.0 | 1.0 | 1.0 | 637868.3 | 1.0 | 1.0 |
| | 0.05 | 1.0 | 35075.4 | 1.0 | 1.0 | 0.99 | 19033.2 | 0.94 | 1.0 | 1.0 | 642238.1 | 1.0 | 1.0 |
| Auer-A1-M | 0.1 | 1.0 | 36231.0 | 1.0 | 1.0 | 0.96 | 31614.2 | 0.76 | 0.99 | 1.0 | 646958.6 | 1.0 | 1.0 |
| | 0.2 | 1.0 | 43351.1 | 0.98 | 1.0 | 0.74 | 47045.5 | 0.22 | 0.87 | 1.0 | 658259.9 | 1.0 | 1.0 |
| | 0.3 | 1.0 | 43722.6 | 0.95 | 1.0 | 0.50 | 23726.7 | 0.03 | 0.71 | 1.0 | 680222.4 | 1.0 | 1.0 |
| | 0.4 | 1.0 | 45058.0 | 0.83 | 1.0 | 0.20 | 6526.8 | 0.0 | 0.59 | 1.0 | 739778.4 | 1.0 | 1.0 |

being a Pareto accurate algorithm. Although the Auer-A1-M method achieves higher scores compared to its mean-based counterpart, these scores remain low, indicating that Auer-A1-M exhibits significant limitations under adversarial conditions.

### 6.3 Experiments on UVA/PADOVA Diabetes Simulator

Managing type 1 diabetes requires adjusting insulin doses based on carbohydrate intake to maintain blood glucose levels within a safe range. Because this method demands specialized training and knowledge, it is prone to errors among patients (Roversi et al., 2020). To simulate this problem, we conduct *in silico* experiments using the University of Virginia (UVa)/PADOVA T1DM simulator (Man et al., 2014) which simulates the blood glucose levels of type 1 diabetes patients for a given meal event, i.e., carbohydrate content of the meal, and an administered external insulin dose. We select 36 combinations of bolus insulin doses and meal events as potential treatment scenarios for a selected patient and assign each of them to an arm. We aim at identifying the combinations that most effectively reduce the duration of time during which blood glucose levels fall below or exceed the safe range under occasional inaccurate carbohydrate counting. We define the safe range as 100-140 mg/dL. When an arm is pulled, the time above the safe range and below the safe range are observed with a small Gaussian noise with $\sigma = 0.025$, which is also used in equation 1. To simulate real-world cases where patients inaccurately count carbohydrates, resulting in the consumption of meals with incorrect carbohydrate content, we deploy an oblivious contamination that changes the meal event of an arm without changing the bolus insulin dose. We standardize the reward distribution so that it has a mean of zero and a variance of one. The choice for the parameters $\alpha, \delta$, and $\bar{t}$ are the same as the SW-LLVM experiment. We report the average results over 100 runs in Table 2. The results indicate that the Auer-A1 exhibits significant limitations under adversarial conditions, in contrast to our algorithm, which aligns with our theoretical predictions and consistently meets the success criteria across various attack probabilities. Specifically, though it takes excessive amounts of samples, it fails to return an accurate and covering Pareto set. Auer-A1-M method manages to return such a Pareto set, but not without requiring almost 2 orders of magnitude more samples than R-PSI.

## 7 Conclusion

We investigated the Pareto set identification problem under adversarial attacks. We proposed a sample-efficient algorithm that returns a predicted set that abides by the accuracy and coverage requirements. We also proved a sample complexity upper bound that matches the adversary-free case lower bound proved in previous studies in terms of accuracy parameter dependence. We further proved a tighter gap-dependent sample complexity bound. The experimental results support our theoretical predictions, demonstrating robustness in adversarial settings. In contrast, multi-objective methods developed for adversary-free environments showed reduced effectiveness against strong adversaries. Even when these methods were extended with median estimators instead of mean estimators—which provided some improvement—they still struggled to maintain effectiveness against strong adversarial conditions. To the best of our knowledge, this is the first study to propose an algorithm with theoretical guarantees that is capable of approximating the Pareto set when the reward samples are corrupted by adversarial attacks with arbitrary contamination values. An interesting future research direction is to investigate the success and the sample complexity of the Pareto set identification for large-scale MO-MAB problems with correlated arms. Employing skewed stochastic processes with separate mean and median parameters could increase the practicality of robust Pareto set identification methods for large-scale MO-MAB problems, as these processes reduce sample complexity by effectively capturing the correlations between the arms.

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

# A   Appendix

## A.1   Derivation of equation 1

Let any $X$ such that $X - \mathbb{E}[X]$ is $\sigma$-subgaussian distributed, be called a $\sigma$-subgaussian variable. Then, for any $q > 0$, we have :

$$\mathbb{P}\big(X - \mathbb{E}[X] \geq q\big) \leq e^{\frac{-q^2}{2\sigma^2}} \, , \tag{9}$$

$$\mathbb{P}\big(X - \mathbb{E}[X] \leq -q\big) \leq e^{\frac{-q^2}{2\sigma^2}} \, . \tag{10}$$

By definition:

$$\mathbb{P}(X < Q_{R,F}(1/2 + t)) \leq 1/2 + t, \ \forall t \in \left(0, \frac{1}{2}\right) \, . \tag{11}$$

Next, we will bound $Q_{R,F}(1/2 + t) - m$ for the three cases given below. Let $q_t := Q_{R,F}(1/2 + t)$. *Case 1:* $\mathbb{E}[X] \leq m \leq q_t$. *Case 2:* $m \leq \mathbb{E}[X] \leq q_t$. *Case 3:* $m \leq q_t \leq \mathbb{E}[X]$.

*Case 1:* Since $q_t - \mathbb{E}[X] > 0$, then, by equation 9 and equation 11:

$$1 - (1/2 + t) \leq \mathbb{P}\big(X \geq q_t\big) = \mathbb{P}\big(X - \mathbb{E}[X] \geq q_t - \mathbb{E}[X]\big)$$
$$\leq e^{\frac{-(q_t - \mathbb{E}[X])^2}{2\sigma^2}} \, .$$

Simplifying above gives:

$$q_t \leq \mathbb{E}[X] + \sqrt{2\sigma^2 \log\left(\frac{1}{1/2 - t}\right)} \ . \tag{12}$$

Since $m \geq \mathbb{E}[X]$:

$$q_t - m \leq q_t - E[X] \leq \sqrt{2\sigma^2 \log\left(\frac{1}{1/2 - t}\right)} \ .$$

*Case 2:* In this case, equation 12 is valid since $q_t \geq \mathbb{E}[X]$. Also, by equation 10, we can bound $\mathbb{E}[X] - m$:

$$1/2 \leq \mathbb{P}(X \leq m) = \mathbb{P}(X - \mathbb{E}[X] \leq m - \mathbb{E}[X]) \leq e^{\frac{-(\mathbb{E}[X]-m)^2}{2\sigma^2}} \ .$$

Therefore, we have:

$$\mathbb{E}[X] - m \leq \sqrt{2\sigma^2 \log 2} \ . \tag{13}$$

Combining equation 13 with equation 12, we obtain:

$$q_t - m = (q_t - \mathbb{E}[X]) + (\mathbb{E}[X] - m) \leq \sqrt{2\sigma^2 \log\left(\frac{1}{1/2 - t}\right)} + \sqrt{2\sigma^2 \log 2} \ .$$

*Case 3:* By equation 13 and by the fact that $q_t \leq \mathbb{E}(X)$:

$$q_t - m \leq \mathbb{E}[X] - m \leq \sqrt{2\sigma^2 \log 2} \ .$$

Combining the results for all three cases, we obtain:

$$q_t - m \leq \sqrt{2\sigma^2 \log\left(\frac{1}{1/2 - t}\right)} + \sqrt{2\sigma^2 \log 2} \ .$$

Following a similar argument, one can obtain the same bound on $m - q_t$, from which the result follows.

### A.2 Proof of Lemma 3

Consider a simple environment with only 2 arms. Suppose that arm 1 is Pareto optimal and arm 2 is such that

$$\forall d \in [M] : m_1^d - m_2^d = 2D - \zeta \ .$$

Adversary can manipulate the experiment so that $\forall d \in [M]$ , $\lim_{n \to \infty} \hat{m}(\tilde{Y}_{1,1}^d, \cdots, \tilde{Y}_{1,n}^d) = m_1^d - D$ and $\lim_{n \to \infty} \hat{m}(\tilde{Y}_{2,1}^d, \cdots, \tilde{Y}_{2,n}^d) = m_2^d + D$ . This implies that: $\exists N_0 : \ \forall N > N_0, \ \forall d \in [M], \ m_1^d - D - \zeta/4 \leq \hat{m}(\tilde{Y}_{1,1}^d, \cdots, \tilde{Y}_{1,N}^d) \leq m_1^d - D + \zeta/4$ and $m_2^d + D - \zeta/4 \leq \hat{m}(\tilde{Y}_{2,1}^d, \cdots, \tilde{Y}_{2,N}^d) \leq m_2^d + D + \zeta/4$ . Therefore, $\forall N > N_0$:

$$\hat{m}(\tilde{Y}_{2,1}^d, \cdots, \tilde{Y}_{2,N}^d) - \hat{m}(\tilde{Y}_{1,1}^d, \cdots, \tilde{Y}_{1,N}^d) \geq \zeta/2 > 0 \ .$$

Hence, we conclude that, even if infinitely many samples are collected from both arms, it is not possible to decide on their optimality based on the empirical medians.

### A.3 Proof of Lemma 4

The total number of samples $N_{\tau_i}$ taken from arm $i$ at the end of the sampling phase $\tau_i$ is given by:

$$N_{\tau_i} = \sum_{\tau=1}^{\tau_i} n_\tau$$

$$= \lceil 2\beta_{\bar{t},\epsilon} \log(\frac{\pi^2 MK}{6\tilde{\delta}}) \rceil + \sum_{\tau=2}^{\tau_i} \left( 1 + \lceil 4\tau\beta_{\bar{t},\epsilon} \log(\frac{\tau}{\tau-1}) + 2\beta_{\bar{t},\epsilon} \log \frac{(\tau-1)^2 MK\pi^2}{6\tilde{\delta}} \rceil \right)$$

$$\geq 2\beta_{\bar{t},\epsilon} \log(\frac{\pi^2 MK}{6\tilde{\delta}}) + \sum_{\tau=2}^{\tau_i} \left( 4\tau\beta_{\bar{t},\epsilon} \log(\frac{\tau}{\tau-1}) + 2\beta_{\bar{t},\epsilon} \log \frac{(\tau-1)^2 MK\pi^2}{6\tilde{\delta}} \right)$$

After simplifying the r.h.s. of the above display, we obtain:

$$N_{\tau_i} \geq 2\tau_i \beta_{\bar{t},\epsilon} \log\left( \frac{\tau_i^2 MK\pi^2}{6\tilde{\delta}} \right) = 2\tau_i \beta_{\bar{t},\epsilon} \log\left( \frac{\delta}{\tilde{\delta}\delta_{\tau_i}} \right) = 2\tau_i \beta_{\bar{t},\epsilon} \log\left( \frac{1}{\tilde{\delta}_{\tau_i}} \right) \ ,$$

where $\delta_{\tau_i} = \frac{6\delta}{\tau_i^2 MK\pi^2}$, $\tilde{\delta}_{\tau_i} = (\delta_{\tau_i}\tilde{\delta})/\delta$ and $\tilde{\delta}$ is given in equations 3 and 4.

Since $N_{\tau_i} \geq 2\tau_i \beta_{\bar{t},\epsilon} \log(\frac{1}{\tilde{\delta}_{\tau_i}})$ and $R$ is a non-decreasing function, we have

$$R\left( h_\epsilon + \sqrt{\frac{1}{\beta_{\bar{t},\epsilon}\tau_i}} \right) \geq R\left( h_\epsilon + \sqrt{\frac{2\log(1/\tilde{\delta}_{\tau_i})}{N_{\tau_i}}} \right) \ .$$

Next, note that $\tilde{\delta}_{\tau_i} = \delta_{\tau_i}/2$ for oblivious and prescient adversary and $\tilde{\delta}_{\tau_i} = \delta_{\tau_i}/3$ for malicious adversary. The inequality above, and Lemmas 1 and 2 imply that $\forall i \in [K]$ and $\forall d \in [M]$:

$$\mathbb{P}\left( \sup_{m_i^d} |\hat{m}_i^d - m_i^d| \geq R\left( h_\epsilon + \sqrt{\frac{1}{\beta_{\bar{t},\epsilon}\tau_i}} \right) \right) \leq \mathbb{P}\left( \sup_{m_i^d} |\hat{m}_i^d - m_i^d| \geq R\left( h_\epsilon + \sqrt{\frac{2\log(1/\tilde{\delta}_{\tau_i})}{N_{\tau_i}}} \right) \right) \leq \delta_{\tau_i} \ .$$

Inserting $U_{\tau_i}$ and $D$ in the left hand side of the inequality above we obtain:

$$\mathbb{P}\left( \sup_{m_i^d} |\hat{m}_i^d - m_i^d| \geq U_{\tau_i} + D \right) \leq \delta_{\tau_i} \ .$$

The result follows by the union bound:

$$\mathbb{P}(E) \geq 1 - \sum_{i\in[K]} \sum_{j\in[M]} \sum_{\tau_i\geq1} \delta_{\tau_i} = 1 - \sum_{i\in[K]} \sum_{j\in[M]} \sum_{\tau_i\geq1} \frac{6\delta}{\tau_i^2 MK\pi^2}$$

$$= 1 - MK \frac{6}{MK\pi^2} \delta \sum_{\tau_i\geq1} \frac{1}{\tau_i^2} = 1 - \delta \ .$$

## A.4  Proof of Theorem 1

In the proof, we assume that event $E$ holds. First, we prove that the Pareto optimal arms are not eliminated in the 'Elimination' step.

**Lemma 5.** *R-PSI does not eliminate Pareto optimal arms in the 'Elimination' step.*

*Proof.* At elimination step, an arm $i$ is eliminated if $\exists j \in S \setminus \{i\} : \hat{m}_i + D + U_i \prec \hat{m}_j - D - U_j$. By Lemma 4, this implies that $m_i = (m_i - D - U_i) + D + U_i \preceq \hat{m}_i + D + U_i \prec \hat{m}_j - D - U_j \preceq (m_j + D + U_j) - D - U_j$. Hence, $m_i \prec m_j$. By definition of Pareto optimality, this is not possible if $i$ is a Pareto optimal arm. Hence, Pareto optimal arms cannot be discarded at the elimination step of R-PSI. □

Next, we prove that R-PSI has an optimality guarantee of $(2D + \alpha)$, i.e., any arm in $P$ is $(2D + \alpha)$-optimal. Before proving this, we state a technical result that will be used in the proof.

**Lemma 6.** *Suppose an arm $i$ is moved to $O_2$ at some sampling phase $t_1$. Then, the following is satisfied for all $t \geq t_1$:*

$$\forall j \in S, \ \exists d_j \in [M] : m_j^{d_j} + D + \alpha > m_i^{d_j} - D \ .$$

*Proof.* Since $i$ is moved to $O_2$ at sampling phase $t_1$, the condition for $O_2$ requires:

$$\nexists j \in S \setminus \{i\} : \hat{m}_j - U_j + \alpha \preceq \hat{m}_i + U_i \ .$$

This implies:

$$\forall j \in S \setminus \{i\}, \ \exists d_j \in [M] : \hat{m}_j^{d_j} - U_j + \alpha > \hat{m}_i^{d_j} + U_i \ .$$

Applying Lemma 4 gives:

$$\forall j \in S \setminus \{i\}, \ \exists d_j \in [M] : \ (m_j^{d_j} + D + U_j) - U_j + \alpha \geq \hat{m}_j^{d_j} - U_j + \alpha > \hat{m}_i^{d_j} + U_i \geq (m_i^{d_j} - D - U_i) + U_i \ .$$

Hence:

$$\forall j \in S, \ \exists d_j \in [M] : m_j^{d_j} + D + \alpha > m_i^{d_j} - D \ .$$

The result follows by noting that $S$ does not admit any new arms over time. ☐

We are now ready to prove the optimality guarantee for the predicted arms.

**Lemma 7.** *$P$ can only contain arms that are $(2D + \alpha)$-optimal.*

*Proof.* Denote the Pareto optimal arms in $S$ at the beginning of sampling phase $t_2$ by $S^{(p)}$ and the Pareto optimal arms in $P$ at the beginning of sampling phase $t_2$ by $P^{(p)}$. We have $P^* = S^{(p)} \cup P^{(p)}$.

Consider $j \in S$ that is moved to $P$ at sampling phase $t_2$. Note that an arm in $S$ needs to be first moved to $O_1$ to end up in $P$. Assume $S^{(p)} \neq \emptyset$ or $S^{(p)} \neq \{j\}$. Then, $j \in S$ moved to $O_1$ implies that:

$$\nexists i \in S^{(p)} \setminus \{j\} : \hat{m}_j - U_j + \alpha \preceq \hat{m}_i + U_i \ ,$$

or equivalently:

$$\forall i \in S^{(p)} \setminus \{j\}, \ \exists d_i \in [M] : \hat{m}_j^{d_i} - U_j + \alpha > \hat{m}_i^{d_i} + U_i \ .$$

By Lemma 4:

$$\forall i \in S^{(p)} \setminus \{j\}, \ \exists d_i \in [M] : \ (m_j^{d_i} + D + U_j) - U_j + \alpha \geq \hat{m}_j^{d_i} - U_j + \alpha > \hat{m}_i^{d_i} + U_i \geq (m_i^{d_i} - D - U_i) + U_i \ .$$

Hence:

$$\forall i \in S^{(p)} \setminus \{j\}, \ \exists d_i \in [M] : m_j^{d_i} + D + \alpha > m_i^{d_i} - D \ . \tag{14}$$

By Definition 4, this implies that $\Delta_{j,i} < 2D + \alpha$ for all $i \in S^{(p)}$.

Assume that $P^{(p)} \neq \emptyset$. We know each arm $i \in P^{(p)}$ must have visited $O_2$ in some sampling phase $t < t_2$ since algorithm did not terminate for any $t < t_2$. Since $j$ is in $S$ at sampling phase $t$, Lemma 6 implies that $\exists d_i \in [M] : m_j^{d_i} + D + \alpha > m_i^{d_i} - D$. Noting that this holds for all $i \in P^{(p)}$, we obtain

$$\forall i \in P^{(p)}, \ \exists d_i \in [M] : m_j^{d_i} + D + \alpha > m_i^{d_i} - D \ . \tag{15}$$

By Definition 4, this implies that $\Delta_{j,i} < 2D + \alpha$ for all $i \in P^{(p)}$. This together with the result after equation 14 implies that $\Delta_j < 2D + \alpha$. Hence, we conclude that if arm $j$ is moved to $P$, it needs to be $(2D + \alpha)$-optimal.

☐

Next, we show that for any Pareto optimal arm that is not returned in $P$, there exists an arm returned in $P$ which is not worse than the Pareto optimal arm more than $2D$ in any objective.

**Lemma 8.** *If a Pareto optimal arm $p^*$ is not returned in $P$ when R-PSI terminates, then there exists an arm $j$ returned in $P$ such that $m_{p^*} - m_j \preceq 2D$ .*

*Proof.* Lemma 5 guarantees that $p^*$ is not eliminated in the 'Elimination' step of the algorithm given in line 17 of Algorithm 1. Therefore, $p^* \notin P$ can happen only if Algorithm 1 enters the 'else' statement in line 28 of its 'Identification' step (happens when $\forall i \in S$, $U_i \leq \alpha/4$) and $p^* \notin O_1$ when this happens. The algorithm terminates and returns $P$ after this.

$p^* \notin O_1$ in the final sampling phase before termination implies that there exists $l_1 \in S \setminus \{p\}$ such that:

$$\hat{m}_{p^*} - U_{\tau_{p^*}} + \alpha \preceq \hat{m}_{l_1} + U_{\tau_{l_1}} . \tag{16}$$

Let $p^* \preceq l_1$ denote the above relation. Also, considering that $U_{\tau_{l_1}}, U_{\tau_{p^*}} \leq \alpha/4$, the above inequality implies:

$$\hat{m}_{p^*} + U_{\tau_{p^*}} \prec \hat{m}_{l_1} - U_{\tau_{l_1}} + \alpha .$$

Hence:

$$\hat{m}_{l_1} - U_{\tau_{l_1}} + \alpha \npreceq \hat{m}_{p^*} + U_{\tau_{p^*}} . \tag{17}$$

We use the notation $l_1 \npreceq p^*$ to denote the expression above.

*Case 1:* Assume that $l_1 \in O_1$, thus it will be returned in $P$. Applying Lemma 4, we get:

$$m_{p^*} - D - 2U_{\tau_{p^*}} + \alpha \preceq \hat{m}_{p^*} - U_{\tau_{p^*}} + \alpha \preceq \hat{m}_{l_1} + U_{\tau_{l_1}} \preceq m_{l_1} + D + 2U_{\tau_{l_1}} ,$$

which implies that $m_{p^*} - m_{l_1} \preceq 2D + 2U_{\tau_{p^*}} + 2U_{\tau_{l_1}} - \alpha \preceq 2D$ since $\alpha \geq 2U_{\tau_{p^*}} + 2U_{\tau_{l_1}}$.

*Case 2:* Assume that $l_1 \notin O_1$. In this case, there exists $l_2 \in S \setminus \{l_1\}$ such that:

$$\hat{m}_{l_1} - U_{\tau_{l_1}} + \alpha \preceq \hat{m}_{l_2} + U_{\tau_{l_2}} . \tag{18}$$

Observe that $l_2 \neq p^*$. To see this assume that $l_2 = p^*$. This will imply

$$\hat{m}_{l_1} - U_{\tau_{l_1}} + \alpha \preceq \hat{m}_{p^*} + U_{\tau_{p^*}} .$$

This is a contradiction since equation 16 implies that

$$\hat{m}_{p^*} - U_{\tau_{p^*}} - 2U_{\tau_{l_1}} + 2\alpha \preceq \hat{m}_{l_1} - U_{\tau_{l_1}} + \alpha \preceq \hat{m}_{p^*} + U_{\tau_{p^*}} \Rightarrow 2\alpha \preceq 2U_{\tau_{l_1}} + 2U_{\tau_{p^*}} ,$$

which does not hold. Thus, we conclude that $l_2 \neq p^*$. Chaining equation 18 with equation 16, we obtain

$$\hat{m}_{p^*} - U_{\tau_{p^*}} + \alpha \preceq \hat{m}_{p^*} - U_{\tau_{p^*}} + \alpha + (\alpha - 2U_{\tau_{l_1}}) \preceq \hat{m}_{l_1} - U_{\tau_{l_1}} + \alpha \preceq \hat{m}_{l_2} + U_{\tau_{l_2}} ,$$

which implies that $m_{p^*} - m_{l_2} \preceq 2D + 2U_{\tau_{p^*}} + 2U_{\tau_{l_2}} - \alpha \preceq 2D$ since $\alpha \geq 2U_{\tau_{p^*}} + 2U_{\tau_{l_2}}$. If $l_2 \in O_1$, we are done. If not, we prove by induction that there is some $j \in O_1$ for which $m_{p^*} - m_j \preceq 2D$.

Now, let $[l_i]_{i=1}^n$ denote a set of arms in $S$ that satisfies the following:

1. $p^* \preceq l_1 \preceq \cdots \preceq l_n$ .

2. There are no repeating arms in the set.

Now, we will prove by induction that

$$\forall z \in \{p^*\} \cup [l_i]_{i=1}^{k-1}, \ l_k \npreceq z \tag{19}$$

for any $k \in [n]$ . We have already proven in equation 17 that when $k = 1$, equation 19 holds. Now, assuming that equation 19 holds for $k - 1$, we show that it also holds for $k$. First, observe that, since $l_{k-1} \preceq l_k$ and since $\alpha > 2U_{\tau_{l_{k-1}}}$ and $\alpha > 2U_{\tau_{l_k}}$ :

$$\hat{m}_{l_{k-1}} + U_{\tau_{l_{k-1}}} \prec \hat{m}_{l_{k-1}} + U_{\tau_{l_{k-1}}} + (\alpha - 2U_{\tau_{l_{k-1}}}) = \hat{m}_{l_{k-1}} - U_{\tau_{l_{k-1}}} + \alpha \preceq \hat{m}_{l_k} + U_{\tau_{l_k}}$$
$$\prec \hat{m}_{l_k} + U_{\tau_{l_k}} + (\alpha - 2U_{\tau_{l_k}}) = \hat{m}_{l_k} - U_{\tau_{l_k}} + \alpha . \tag{20}$$

Therefore, $\hat{m}_{l_{k-1}} + U_{\tau_{l_{k-1}}} \prec \hat{m}_{l_k} - U_{\tau_{l_k}} + \alpha$ , which implies that $\hat{m}_{l_k} - U_{\tau_{l_k}} + \alpha \not\preceq \hat{m}_{l_{k-1}} + U_{\tau_{l_{k-1}}}$ , or $l_k \not\preceq l_{k-1}$. Also, by assumption, $l_{k-1} \not\preceq z$, $\forall z \in \{p^*\} \cup [l_i]_{i=1}^{k-2}$, or equivalently:

$$\hat{m}_{l_{k-1}} - U_{\tau_{l_{k-1}}} + \alpha \not\preceq \hat{m}_z + U_{\tau_z}, \quad \forall z \in \{p^*\} \cup [l_i]_{i=1}^{k-2} . \tag{21}$$

Also it is shown in equation 20 that:

$$\hat{m}_{l_{k-1}} - U_{\tau_{l_{k-1}}} + \alpha \prec \hat{m}_{l_k} - U_{\tau_{l_k}} + \alpha . \tag{22}$$

Hence, by equation 21, and equation 22:

$$\hat{m}_{l_k} - U_{\tau_{l_k}} + \alpha \not\preceq \hat{m}_z + U_{\tau_z}, \quad \forall z \in \{p^*\} \cup [l_i]_{i=1}^{k-2} .$$

Also, we had proved above that $l_k \not\preceq l_{k-1}$. From this and above, we conclude:

$$\forall z \in \{p^*\} \cup [l_i]_{i=1}^{k-2} \cup \{l_{k-1}\}, \ l_k \not\preceq z$$

which proves the induction. Now, suppose that we can find another arm $l_{n+1}$ that we can add to this set. Since the induction proven above applies to this arm as well, we can deduce that $l_{n+1}$ is not prevented by any other arm in this set to enter $O_1$. Now, suppose that we sequentially keep adding other arms to this set. Considering that we have a finite amount of arms in $S$, we will eventually reach a point where we will not be able to add any more arms. Denote by $L$ a set that is constructed through such a procedure and denote the last element of this set by $j$. Then, $j$ satisfies the following:

$$p^* \preceq l_1 \cdots \preceq j .$$

Also, since we cannot keep adding any more arms to this set, we must have:

$$\forall k \in S \setminus (L \cup \{p^*\}) : j \not\preceq k .$$

Considering that $j$ satisfies the above proven induction:

$$\forall k \in L \cup \{p^*\} : j \not\preceq k .$$

Hence, combining two inequalities above yields:

$$\forall k \in S : j \not\preceq k .$$

The above inequality means that arm $j$ is returned in $P$. Next, we prove by another induction that $p^* \preceq l_k$ for any $l_k \in L$ . This is already given for the first element, i.e., $p^* \preceq l_1$ . Now assume that this is true for arm $l_{k-1}$ . Then, $p^* \preceq l_{k-1}$ or equivalently $\hat{m}_{p^*} - U_{\tau_{p^*}} + \alpha \preceq \hat{m}_{l_{k-1}} + U_{\tau_{l_{k-1}}}$. As shown in equation 20, we also have that:

$$\hat{m}_{l_{k-1}} + U_{\tau_{l_{k-1}}} \prec \hat{m}_{l_k} + U_{\tau_{l_k}} .$$

Hence:

$$\hat{m}_{p^*} - U_{\tau_{p^*}} + \alpha \preceq \hat{m}_{l_k} + U_{\tau_{l_k}}$$

or $p^* \preceq l_k$ and the induction is proven. Since arm $j$ described above is part of this set, $p^* \preceq j$ or equivalently:

$$\hat{m}_{p^*} - U_{\tau_{p^*}} + \alpha \preceq \hat{m}_j + U_{\tau_j} .$$

Applying Lemma 4 above, we get:

$$m_{p^*} - D - 2U_{\tau_{p^*}} + \alpha \preceq \hat{m}_{p^*} - U_{\tau_{p^*}} + \alpha \preceq \hat{m}_j + U_{\tau_j} \preceq m_j + D + 2U_{\tau_j} .$$

Hence, $m_{p^*} - D - 2U_{\tau_{p^*}} + \alpha \preceq m_j + D + 2U_{\tau_j}$ . Since $\alpha \geq 2U_{\tau_{p^*}} + 2U_{\tau_j}$, this implies that $m_{p^*} - D \preceq m_j + D$. $\quad\square$

Next, we state the termination condition for R-PSI in the remark below. We obtain this condition based on the observation that if the algorithm enters the "else" statement inside the identification step at some round $t$, then it terminates after performing the operation inside this "else".

**Remark 3.** *R-PSI terminates at the latest when $\forall i \in S, U_i \leq \alpha/4$.*

Next, we give a relaxed elimination condition for the arms that are $(4D+\alpha)$-suboptimal in the lemma below. The proof technique of this lemma is similar to the previous lemmas and can be found in Appendix A.7.

**Lemma 9.** *An arm $i$ that is $(4D + \alpha)$-suboptimal is guaranteed to be eliminated at the latest when $\forall k \in S, U_k < \bar{\Delta}_i/4$ where $\bar{\Delta}_i = \Delta_i - 4D$ .*

With this, we are ready to complete the proof of Theorem 1 .

*Final Steps in the Proof of Theorem 1*: By Remark 1 and 3, $\forall i \in [K]$, we have:

$$\tau_i \leq \inf\{\tau : U_\tau \leq \alpha/4\} \leq \inf\{\tau : U_\tau \leq \alpha/5\} \ . \tag{23}$$

By Lemma 9, we can obtain tighter bounds on the number of sampling rounds of arms that are $(4D + \alpha)$-suboptimal:

$$\tau_i \leq \inf\{\tau : U_\tau < \bar{\Delta}_i/4\} \leq \inf\{\tau : U_\tau \leq \bar{\Delta}_i/5\} \ . \tag{24}$$

The total number of samples taken from an arm that has been selected for $\tau$ sampling phases can be bounded by:

$$N_\tau = n_0 + \sum_{\tilde{\tau}=2}^{\tau} n_{\tilde{\tau}}$$

$$= \lceil 2\beta_{\bar{t},\epsilon}\log(\frac{\pi^2 MK}{6\tilde{\delta}})\rceil + \sum_{\tilde{\tau}=2}^{\tau} 1 + \lceil 4\tilde{\tau}\beta_{\bar{t},\epsilon}\log(\frac{\tilde{\tau}}{\tilde{\tau}-1}) + 2\beta_{\bar{t},\epsilon}\log\frac{(\tilde{\tau}-1)^2 MK\pi^2}{6\tilde{\delta}}\rceil$$

$$\leq 1 + 2\beta_{\bar{t},\epsilon}\log(\frac{\pi^2 MK}{6\tilde{\delta}}) + \sum_{\tilde{\tau}=2}^{\tau}\left(2 + 4\tilde{\tau}\beta_{\bar{t},\epsilon}\log(\frac{\tilde{\tau}}{\tilde{\tau}-1}) + 2\beta_{\bar{t},\epsilon}\log\frac{(\tilde{\tau}-1)^2 MK\pi^2}{6\tilde{\delta}}\right) \ .$$

Simplifying r.h.s. of the above display, we obtain:

$$N_\tau \leq 2\tau(\beta_{\bar{t},\epsilon}\log(\frac{\tau^2 MK\pi^2}{6\tilde{\delta}}) + 1) \ .$$

Now, we can bound the sample complexity as follows.

$$N = \sum_{i=1}^{K} N_{\tau_i}$$

$$= \sum_{i: \Delta_i > 4D+\alpha} N_{\tau_i} + \sum_{i: \Delta_i \leq 4D+\alpha} N_{\tau_i}$$

$$\leq \sum_{i: \Delta_i > 4D+\alpha} 2\tau_{(\bar{\Delta}_i)}\left(\beta_{\bar{t},\epsilon}\log\left(\frac{\tau_{(\bar{\Delta}_i)}^2 MK\pi^2}{6\tilde{\delta}}\right) + 1\right) + \sum_{i: \Delta_i \leq 4D+\alpha} 2\tau_{(\alpha)}\left(\beta_{\bar{t},\epsilon}\log\left(\frac{\tau_{(\alpha)}^2 MK\pi^2}{6\tilde{\delta}}\right) + 1\right)$$

$$= \sum_{i=1}^{K} 2\tau_{(\Delta_i^\alpha)}\left(\beta_{\bar{t},\epsilon}\log\left(\frac{\tau_{(\Delta_i^\alpha)}^2 MK\pi^2}{6\tilde{\delta}}\right) + 1\right) \tag{25}$$

$$\leq K\tau_{(\alpha)}\left(2\beta_{\bar{t},\epsilon}\log\left(\frac{\tau_{(\alpha)}^2 MK\pi^2}{6\tilde{\delta}}\right) + 2\right) \ ,$$

where $\Delta_i^\alpha := \max(\alpha, \bar{\Delta}_i)$.

Lastly, the Pareto accuracy of R-PSI follows from Lemma 7 and 8. $\qquad\square$

## A.5    Proof of Corollary 1

Let $\tau_- := \tau_{(\alpha)} - 1$ . By definition of $\tau_{(\alpha)}$, we have:

$$U_{\tau_\alpha - 1} = U_{\tau_-} = R\left(h_\epsilon + 1/\sqrt{\beta_{\bar{t},\epsilon}\tau_-}\right) - R(h_\epsilon) > \alpha/5 \ .$$

Using $R(t)$ from Example 1, we have:

$$\sigma\sqrt{2}\left(\sqrt{\log\left(\frac{1}{1/2 - h_\epsilon - \sqrt{\frac{1}{\beta_{\bar{t},\epsilon}\tau_-}}}\right)} - \sqrt{\log\left(\frac{1}{1/2 - h_\epsilon}\right)}\right) > \alpha/5$$

$$\Leftrightarrow 2\sigma^2 \log\left(\frac{1}{1/2 - h_\epsilon - \sqrt{\frac{1}{\beta_{\bar{t},\epsilon}\tau_-}}}\right) > \frac{\alpha^2}{25} + 2\sigma^2 \log\left(\frac{1}{1/2 - h_\epsilon}\right) + \frac{2\alpha\sigma\sqrt{2}}{5}\sqrt{\log\left(\frac{1}{1/2 - h_\epsilon}\right)}$$

$$\Leftrightarrow \log\left(\frac{1}{1/2 - h_\epsilon - \sqrt{\frac{1}{\beta_{\bar{t},\epsilon}\tau_-}}}\right) > \frac{\alpha^2}{50\sigma^2} + \log\left(\frac{1}{1/2 - h_\epsilon}\right) + \frac{\alpha\sqrt{2}}{5\sigma}\sqrt{\log\left(\frac{1}{1/2 - h_\epsilon}\right)} \ .$$

Let $c_1 = \frac{1}{50\sigma^2}$ and $c_2 = \frac{\sqrt{2}}{5\sigma}\sqrt{\log\left(\frac{1}{1/2 - h_\epsilon}\right)}$. Continuing from the above display,

$$\log\left(\frac{1}{1/2 - h_\epsilon - \sqrt{\frac{1}{\beta_{\bar{t},\epsilon}\tau_-}}}\right) > \alpha^2 c_1 + \log\left(\frac{1}{1/2 - h_\epsilon}\right) + \alpha c_2$$

$$\Leftrightarrow \frac{1}{1/2 - h_\epsilon - \sqrt{\frac{1}{\beta_{\bar{t},\epsilon}\tau_-}}} > \exp\left(\alpha^2 c_1 + \log\left(\frac{1}{1/2 - h_\epsilon}\right) + \alpha c_2\right)$$

$$\Leftrightarrow \frac{1}{\exp\left(\alpha^2 c_1 + \log\left(\frac{1}{1/2 - h_\epsilon}\right) + \alpha c_2\right)} > 1/2 - h_\epsilon - \sqrt{\frac{1}{\beta_{\bar{t},\epsilon}\tau_-}}$$

$$\Leftrightarrow \sqrt{\frac{1}{\beta_{\bar{t},\epsilon}\tau_-}} > 1/2 - h_\epsilon - \exp\left(-\alpha^2 c_1 - \log\left(\frac{1}{1/2 - h_\epsilon}\right) - \alpha c_2\right)$$

$$\Leftrightarrow \sqrt{\frac{1}{\beta_{\bar{t},\epsilon}\tau_-}} > (1/2 - h_\epsilon)(1 - \exp(-\alpha^2 c_1 - \alpha c_2))$$

$$\Leftrightarrow \tau_- < \frac{1}{(1/2 - h_\epsilon)^2 \beta_{\bar{t},\epsilon}}\left(\frac{1}{1 - \exp(-\alpha^2 c_1 - \alpha c_2)}\right)^2 \ .$$

Continuing from the above display, using $\frac{1}{1 - e^{-x}} = 1 + \frac{1}{e^x - 1} \leq 1 + \frac{1}{x}$ we get:

$$\tau_- < \frac{1}{(1/2 - h_\epsilon)^2 \beta_{\bar{t},\epsilon}}\left(1 + \frac{1}{\alpha^2 c_1 + \alpha c_2}\right)^2 \ .$$

Recall that $\tau_- := \tau_{(\alpha)} - 1$:

$$\tau_{(\alpha)} < 1 + \frac{1}{(1/2 - h_\epsilon)^2 \beta_{\bar{t},\epsilon}}\left(1 + \frac{1}{\alpha^2 c_1 + \alpha c_2}\right)^2$$

$$= 1 + \frac{(\bar{t} - h_\epsilon)^2}{(1/2 - h_\epsilon)^2}\left(1 + \frac{1}{\alpha^2 c_1 + \alpha c_2}\right)^2$$

$$< 1 + \left(1 + \frac{1}{\alpha^2 c_1 + \alpha c_2}\right)^2 \tag{26}$$

$$\leq 1 + \left(1 + \frac{1}{\alpha c_2}\right)^2$$

$$\leq \left(\sqrt{2} + \frac{1}{\alpha c_2}\right)^2$$

$$\leq \left(\frac{2}{\alpha c_2}\right)^2 \tag{27}$$

$$= \frac{4}{\alpha^2 c_2^2}$$

$$= \frac{50\sigma^2}{\alpha^2 \log\left(\frac{1}{1/2 - h_\epsilon}\right)} \; , \tag{28}$$

where equation 27 holds when $\alpha c_2 \leq 1/\sqrt{2}$, which is valid when $\alpha \to 0$ and $\epsilon$ is fixed. Equation 26 holds due to $(\bar{t} - h_\epsilon)^2 < (1/2 - h_\epsilon)^2$ which follows from

$$(\bar{t} - h_\epsilon)^2 - (1/2 - h_\epsilon)^2 = (\bar{t} - 1/2)(\bar{t} - 2h_\epsilon + 1/2) < 0 \; .$$

The last line holds due to $h_\epsilon \leq \bar{t} < 1/2$. For oblivious and prescient adversaries, Theorem 1 has the assumption $\epsilon \leq \frac{2\bar{t}}{1+2\bar{t}}$ which is equivalent to $\frac{\epsilon}{2(1-\epsilon)} \leq \bar{t}$. For malicious adversary, Theorem 1 has the assumption $\epsilon \leq \bar{t}$. Hence, $h_\epsilon \leq \bar{t}$ for all adversaries. By Definition 2, $\bar{t} \in (0, 1/2)$. Hence, $h_\epsilon \leq \bar{t} < 1/2$.

Starting from equation 7:

$$N \leq K\tau_{(\alpha)}\left(2\beta_{\bar{t},\epsilon} \log\left(\frac{\tau_{(\alpha)}^2 MK\pi^2}{6\tilde{\delta}}\right) + 2\right)$$

$$\leq 4K\tau_{(\alpha)}\beta_{\bar{t},\epsilon} \log\left(\frac{\tau_{(\alpha)}^2 MK\pi^2}{6\tilde{\delta}}\right) \tag{29}$$

$$= 4K\tau_{(\alpha)}\beta_{\bar{t},\epsilon}\left(\log\left(\frac{\tau_{(\alpha)}^2 MK}{\tilde{\delta}}\right) + \log\left(\frac{\pi^2}{6}\right)\right)$$

$$\leq 8K\tau_{(\alpha)}\beta_{\bar{t},\epsilon} \log\left(\frac{\tau_{(\alpha)}^2 MK}{\tilde{\delta}}\right) \tag{30}$$

$$\leq 8K\tau_{(\alpha)}\beta_{\bar{t},\epsilon} \log\left(\frac{\tau_{(\alpha)}^2 M^2 K^2}{\tilde{\delta}^2}\right)$$

$$= 16K\tau_{(\alpha)}\beta_{\bar{t},\epsilon} \log\left(\frac{\tau_{(\alpha)} MK}{\tilde{\delta}}\right) \; .$$

Equations 29 and 30 hold for large $M, K$ and small $\tilde{\delta}$ values. Plugging in equation 28, we have

$$N \leq 16K \frac{50\sigma^2}{\alpha^2 \log\left(\frac{1}{1/2 - h_\epsilon}\right)} \beta_{\bar{t},\epsilon} \log\left(\frac{\frac{50\sigma^2}{\alpha^2 \log\left(\frac{1}{1/2 - h_\epsilon}\right)} MK}{\tilde{\delta}}\right)$$

$$= 800 \frac{K\sigma^2}{\alpha^2 \log\left(\frac{1}{1/2 - h_\epsilon}\right)} \beta_{\bar{t},\epsilon} \log\left(\frac{50\sigma^2 MK}{\alpha^2 \log\left(\frac{1}{1/2 - h_\epsilon}\right)\tilde{\delta}}\right) \tag{31}$$

$$= \mathcal{O} \left( \frac{\sigma^2 K \beta_{\bar{t},\epsilon}}{\alpha^2 \log\left(\frac{1}{1/2-h_\epsilon}\right)} \log\left( \frac{\sigma^2 M K}{\alpha^2 \log\left(\frac{1}{1/2-h_\epsilon}\right)\tilde{\delta}} \right) \right) .$$

Let us analyze the bound under the assumption $\epsilon = 0$ and $\bar{t}$ is a constant. We have $\log\left(\frac{1}{1/2-h_\epsilon}\right) = \log(2)$, $\beta_{\bar{t},\epsilon} = \frac{1}{\bar{t}^2}$. Starting from equation 31, we have

$$N \le 800 \frac{\sigma^2 K \beta_{\bar{t},\epsilon}}{\alpha^2 \log\left(\frac{1}{1/2-h_\epsilon}\right)} \log\left( \frac{50\sigma^2 M K}{\alpha^2 \log\left(\frac{1}{1/2-h_\epsilon}\right)\tilde{\delta}} \right)$$

$$= \frac{800\sigma^2}{\bar{t}^2 \log(2)} \frac{K}{\alpha^2} \left( \log\left( \frac{50\sigma^2 M K}{\alpha^2 \tilde{\delta} \log(2)} \right) \right)$$

$$= \mathcal{O}\left( \frac{K\sigma^2}{\alpha^2} \log\left( \frac{\sigma^2 M K}{\alpha\tilde{\delta}} \right) \right) .$$

## A.6 Proof of Corollary 2

Let $\tau_- := \tau_{\left(\Delta_i^\alpha\right)} - 1$. By definition of $\tau_{\left(\Delta_i^\alpha\right)}$, we have:

$$U_{\tau_-} = R\left( h_\epsilon + 1/\sqrt{\beta_{\bar{t},\epsilon}(\tau_-)} \right) - R(h_\epsilon) > (\Delta_i^\alpha)/5 .$$

Let us define $z := B\bar{m}_2$ for brevity. Notice that $z > 0$ since $B > 0$ by Definition 3 and $\bar{m}_2 > 0$ as it is the maximum over the median of non-negative values. Also, $\beta_{\bar{t},\epsilon} > 0$ for oblivious, prescient and malicious adversaries. Using $R(t) = B\bar{m}_2 t$, we have

$$z\left( h_\epsilon + (\beta_{\bar{t},\epsilon}\tau_-)^{\frac{-1}{2}} \right) - zh_\epsilon > \frac{(\Delta_i^\alpha)}{5} \Leftrightarrow (\beta_{\bar{t},\epsilon}\tau_-)^{\frac{-1}{2}} z > \frac{(\Delta_i^\alpha)}{5} \Leftrightarrow (\beta_{\bar{t},\epsilon}\tau_-)^{\frac{-1}{2}} > \frac{(\Delta_i^\alpha)}{5z} \Leftrightarrow (\beta_{\bar{t},\epsilon}\tau_-)^{-1} > \frac{(\Delta_i^\alpha)^2}{25z^2}$$

$$\Leftrightarrow \beta_{\bar{t},\epsilon}\tau_- < \frac{25z^2}{(\Delta_i^\alpha)^2} \Leftrightarrow \tau_- < \frac{25z^2}{\beta_{\bar{t},\epsilon}(\Delta_i^\alpha)^2} \Leftrightarrow \tau_{\left(\Delta_i^\alpha\right)} < 1 + \frac{25z^2}{\beta_{\bar{t},\epsilon}(\Delta_i^\alpha)^2} .$$

Recall equation 25:

$$N \le \sum_{i=1}^{K} 2\tau_{\left(\Delta_i^\alpha\right)} \left( \beta_{\bar{t},\epsilon} \log\left( \frac{\tau_{\left(\Delta_i^\alpha\right)}^2 M K \pi^2}{6\tilde{\delta}} \right) + 1 \right) .$$

Putting them together,

$$N \le \sum_i^K 2\left( \beta_{\bar{t},\epsilon} + \frac{25\bar{m}_2^2 B^2}{(\Delta_i^\alpha)^2} \right) \left( \log\left( \frac{\left(1 + \frac{25\bar{m}_2^2 B^2}{\beta_{\bar{t},\epsilon}(\Delta_i^\alpha)^2}\right)^2 M K \pi^2}{6\tilde{\delta}} \right) + 1 \right) .$$

## A.7 Proof of Lemma 9

First, we give the following lemma needed in the proof.

**Lemma 10.** *Consider an arm $i$ that is $(2D + \alpha)$-suboptimal and suppose that the Pareto optimal arm $j = \arg\max_{z \in P^*} \Delta_{i,z}$ is moved to $P$ at $t_0$ before the termination round. Then $i$ is eliminated at $t \le t_0$.*

*Proof.* Assume that $i \in S$ at the beginning of 'Identification' step at sampling phase $t_0$. By Lemma 6, $\exists d_i \in [M] : m_i^{d_i} + (2D + \alpha) > m_j^{d_i}$, implying $\Delta_i < 2D + \alpha$, hence arm $i$ cannot be $(2D + \alpha)$-suboptimal. Therefore, it is not possible for arm $i$ to be in $S$ after the 'Elimination' step at $t_0$. Similarly, $i$ cannot be in $P$ at $t_0$ because of Lemma 7. These together imply that it was eliminated at a round $t \le t_0$. □

Next we proceed proving Lemma 9. By Lemma 7, if $i \notin S$, then $i$ is already eliminated since $i$ cannot be in $P$. Now, suppose that $i \in S$. Consider the optimal arm $j = \arg\max_{k \in P^*} \Delta_{i,k}$. Note that $\Delta_{i,j} = \Delta_i > 4D + \alpha$ and

$$\forall d \in [M], \ m_j^d \ge m_i^d + \Delta_i \ . \tag{32}$$

Now, suppose that $j \in S$ so that $U_j \le \bar{\Delta}_i/4$. Then, by Lemma 4 and by equation 32:

$$\begin{aligned}\hat{m}_j^d - D - U_j &\ge m_j^d - 2D - 2U_j \ge m_i^d + \Delta_i - 2D - 2U_j \\ &\ge \hat{m}_i^d + \Delta_i - 3D - 2U_j - U_i \ . \end{aligned} \tag{33}$$

Considering that $\Delta_i > 4D$ and $U_i, U_j < \bar{\Delta}_i/4$:

$$\hat{m}_i^d + \Delta_i - 3D - 2U_j - U_i = \hat{m}_i^d + \Delta_i - 4D - 2U_j - 2U_i + D + U_i > \hat{m}_i^d + D + U_i \ . \tag{34}$$

We conclude from equation 33 and equation 34:

$$\hat{m}_j^d - D - U_j > \hat{m}_i^d + D + U_i, \ \forall d \in [M] \ .$$

Hence, by the elimination rule of R-PSI, when $j \in S$, $i$ is eliminated. Now suppose that $j \notin S$. Since $j$ cannot be eliminated by Lemma 5, $j \in P$. By Lemma 10, $i$ is eliminated.

Lastly, suppose that at the earliest round where $\forall k \in S$, $U_k \le \bar{\Delta}_i/4$, the algorithm enters the else statement in the 'Identification' step. Then, the algorithm terminates by Remark 3 and $i$ is not included in $P$ by Lemma 7. To summarize, in all possible scenarios, we showed that $i$ is guaranteed to be eliminated (if it is not already eliminated) as soon as $\forall k \in S$, $U_k \le \bar{\Delta}_i/4$.

