# OpenReview forum: "Robust Pareto Set Identification With Contaminated Bandit Feedback"
_TMLR — Rejected by TMLR_

### Review · Reviewer_AJZo · 2024-08-28

**Summary Of Contributions:**

This study addresses the problem of identifying the Pareto set in multi-objective multi-armed bandits (MO-MAB) when rewards are contaminated by an adversary. With a fixed probability, true rewards are replaced by samples from a contamination distribution. The work proposes a robust, sample median-based multi-objective adaptive elimination algorithm that returns an approximately correct Pareto set.

**Audience:**

No

**Claims And Evidence:**

Yes

**Requested Changes:**

1. The function $R(\cdot)$ is not specifically provided before Lemma 1 and 2. Do you mean the function $R$ that appears in Lemma 1 and 2 is exactly example 1? or it can be any non-decreasing function?  one of  $C_{\bar{t},R}$ and $C_{R, \bar{t}}$ should be a typo in Lemma 1,2 and Definition 2.

2. It is better to replace the baseline with other median-based algorithms, e.g. an adaption of Algorithm 2 of Altschule et al. to multiple objectives.

3. If any, the authors could enhance the manuscript by clearly delineating their contributions, particularly with respect to the development of proof concepts or algorithmic idea.

**Strengths And Weaknesses:**

**Strengths**
1. The paper is overall well-written and concise.
2. The theoretical claim is solid and well-supported by the previous work of Altschule et al.
3. The experiments run sufficiently on multiple datasets.

**Weaknesses**
1. Median-based analysis seems to be a niche. It would be valuable to provide some motivational examples of why we consider the median instead of the mean, especially in the case of multiple objectives.

2. The main investigation of this paper is to propose Algorithm 1 and theoretically provide its upper bound. However, there is no tightness analysis of the proposed upper bound. Even when compared with a less tight lower bound (if any), the insights gained can be very valuable. Hence, the significance of the upper bound is unclear in its current form.

3. The technical contributions of this work are quite limited. The concentration theorems closely mirror those presented in the prior work by Altschuler et al., and the algorithmic design bears a significant resemblance to Algorithm 3 from their research.

4. In the experimental part, the choice of baseline is not suitable.  The algorithm of Auer et al. (2016) is not based on contamination. More importantly, Auer et al. (2016) consider the mean-based Pareto set instead of the median-based one. It would be valuable to compare with a baseline designed for the median-based Pareto set, e.g. an adaption of Algorithm 2 of Altschule et al. to multiple objectives.

---

> ### Author Response · Authors · 2024-09-27
> **Rebuttal**
>
> We thank the reviewer for their thoughtful comments and valuable insights.
>
> # Answer to Weakness 1
>
> The choice of median-based estimators is necessary due to the nature of contamination in our setting, where contamination is unbounded in amplitude but bounded in frequency.  Under this contamination model, mean estimators cannot reliably provide accurate estimations of the true mean values. To further illustrate this, we can provide additional motivational examples of scenarios with high contamination amplitudes. Such situations often arise in sensor networks in harsh environments. In a sensor network that is monitoring multiple environmental conditions (e.g., temperature, humidity), occasional faulty sensors can produce readings that are wildly different from the true values due to hardware malfunctions or extreme environmental conditions. Another example is marketing campaigns: A company wants to maximize both brand awareness and conversion rates. Occasionally, data on campaign reach or sales might be distorted by external factors like fake social media accounts, leading to skewed metrics for both objectives. Hence, optimizing multiple conditions under contamination requires median statistics.
>
> # Answer to Weakness 2
>
> To analyze the tightness of the upper bounds, we consider specific problem instances. Under these problem instances, the success condition given in Definition 5 of our work implies success conditions of the previous works. This further implies that our method (and also other Pareto accurate methods) is also bound to the lower bounds established in the mentioned works. A discussion of this analysis can be found in Remark 2 of the revised document, and we also provide a detailed discussion below.
>
> To analyze the tightness of the upper bounds, we consider a specific problem instance where $M=1$. In this case, $P^*=\arg \max \_i m\_i$ and $\Delta\_i=\max \_j m\_j-m\_i$. Thus, the accuracy condition for Pareto accurate algorithms given in Definition 5 reduces to $\forall i \in P, m\_i \geq \max \_j m\_j-2 D-\alpha$. The coverage condition reduces to $\forall j \in P^*, \exists i \in P: m\_j-m\_i \leq 2 D$. Success conditions considered in Altschuler et al. (2019) require that any successful algorithm, for any $\alpha \geq 0$, should return a single arm $I$ such that $m\_{i^*}-m\_I-U\_{i^*}-U_I \leq \alpha$, where $i^*=\arg \max \_i m\_i$. In terms of our notation, this condition is $m\_I \geq \max \_j m\_j-2 D-\alpha$, which is equivalent to our accuracy condition. Thus, we can say that Altschuler et al. (2019) studies a success condition that is weaker than ours. Any lower bound for the success condition of Altschuler et al. (2019) also holds in our case. In particular, the lower bound stated in Theorem 18 of Altschuler et al. (2019) holds, which is given by $\Omega \left( \sum_{i \in [K] \setminus \{i^*\} } \frac{1}{{\max(\Delta}_i - 2D, \alpha)^2} \log (\frac{1}{\delta}) \right)$. In terms of dependence on $\alpha$, both Corollary 1 and 2 match this lower bound up to logarithmic terms.
>
> Let us consider another specific problem instance where the reward distributions are from family of distributions $\mathcal{F}\_{B, \bar{t}}$. Then, when $\epsilon =0$, we have $D = 0$ for all adversaries. Thus, the accuracy condition for Pareto accurate algorithms given in Definition 5 reduces to $\forall i \in P, \Delta\_i \leq \alpha$. The coverage condition reduces to $\forall j \in P^*, \exists i \in P: m\_j-m\_i \preceq 0$. However, by definition of a Pareto optimal arm $j$, such an arm $i$ cannot exist. Therefore, our coverage condition implies that all Pareto optimal arms must be in $P$, i.e., $P^* \subseteq P$. Thus, our coverage condition is equivalent to $P^* \subseteq P$. These two conditions are the same as success condition 1.a and 1.b of Auer et al. (2016), which is, $\forall i \in P: \Delta\_i^* \leq \alpha$ and $P^* \subseteq P$ respectively. Note that in Auer et al. (2016), $\Delta\_i^*=\max \_{j \in P^*} m(i, j)=\max (\{0, \min \_d(y\_j^d-y\_i^d)\})$ where $y\_i^d$ represents the mean as opposed to the median. Also, their Pareto set definition is in terms of means instead of medians. So their gaps are in terms of means, while our gaps are in terms of medians. However, when the distinction between using medians and means is ignored, our upper bounds match their lower bound up to logarithmic terms in terms of $\alpha$ dependency.

---

> ### Author Response · Authors · 2024-09-27
> **Rebuttal**
>
> # Answer to Weakness 3
> While there may appear to be similarities between Algorithm 3 of Altschuler et al. (2019) and our R-PSI, there are significant differences that distinguish our work. R-PSI samples the least sampled design each round, unlike Algorithm 3, which samples every design. Although both algorithms share similar elimination steps, R-PSI includes an intricate Pareto identification phase where Pareto optimal points that are needed to establish the status of other points they may dominate, are not immediately removed from the set of active designs. Perhaps most importantly, the termination condition in Algorithm 3 is based on having only one design left, which does not extend to the multi-objective setting. This is because number of optimal arms cannot be known beforehand. Furthermore, estimating the number of optimal arms and stopping when the estimation is reached is not viable, since if the estimation undershoots, the algorithm would keep taking samples and never terminate. And if it overshoots, then it may return a suboptimal arm. In contrast, R-PSI’s termination condition is based on R-specific (Reward function-specific) certainty and hence can be used in practice. R-PSI makes solving the contaminated MO-MAB problem feasible by addressing these termination issues.
>
> We acknowledge that the concentration theorems are similar to those of Altschuler et al. (2019). However, the most notable theoretical contributions of our work stem from other novelties. Example 1 given in Section 3.2 is an important material. By providing $R(\cdot)$ for subgaussian distributions, we significantly broaden the scope of the concentration theorems. Subgaussian distributions encompass a wide family of distributions commonly encountered in practice. This is important for improving the practical utility of the theoretical results, enabling their application to a wider array of real-world problems where reward distributions are often subgaussian. Moreover, the set operations of R-PSI allows it to overcome the mentioned termination condition issue, making it a feasible contaminated MO-MAB method whereas no straightforward extension of prior works achieve feasibility. These set operations are intricate, thus proving that R-PSI returns an ($\epsilon$,$\delta$)-PAC Pareto set requires a rigorous analysis (see Lemmata 5, 6, 7, 8). We extend the contaminated BAI problem to multi-objectives and provide sample complexity bounds for R-PSI on a very large family of problems (Theorem 1, Corollary 2). Furthermore, we extend previous results to $\sigma$-subgaussian reward distributions (by using Example 1) and provide sample complexity bounds of R-PSI (Corollary 1). We also discuss the tightness of Corollary 1 and 2 in Remark 2 of the revised document.
> # Answer to Weakness 4
> We kindly refer the reviewer to the response to the requested change where we discuss the problems related to the feasibility of the adaptions of Algorithm 2 of Altschuler et al to the MO-MAB problem. Instead of comparing with Altschuler et al., in the revised manuscript, we included numerical comparisons with a median-based adaptation of the Algorithm 1 (Auer-A1-M) in Auer et al. (2016). New results can be found under the numerical results section. Overall, our algorithm achieves higher performance.
> # Answer to Requested Change 1
> Indeed, the function $R$ (in Lemma 1 and 2) can be any positive, non-decreasing function defined on domain $[0,t]$. To ensure clarity, we have now added a sentence in both Lemma 1 and 2 that emphasizes this point. Also, the typo is fixed.

---

> ### Author Response · Authors · 2024-09-27
> **Rebuttal**
>
> # Answer to Requested Change 2
>
> To address this issue, we extended Algorithm 2 of Altschuler et al. (2019) to contaminated and multi-objective setting for family of distributions $C_{R, \bar{t}}$ for any $R$ from Definition 2. Altschuler's Algorithm 2 operates on the principle of elimination, where designs that are not the best arm are progressively removed. The process ensures that suboptimal arms are eliminated as soon as the algorithm is sufficiently confident about the accuracy of the median estimates. Given enough samples, all suboptimal arms are eventually discarded, leaving only the optimal arm as the final output. However, the termination condition in Altschuler's Algorithm 2 is based on having only one design left, which does not extend to the multi-objective setting. This is because number of Pareto optimal arms cannot be known beforehand. Furthermore, estimating or guessing the number of Pareto optimal arms and stopping when the estimation is reached is not viable, since if the estimation undershoots, the algorithm would keep taking samples and never terminate. And if it overshoots, then it may return a suboptimal arm.
>
> Hence, this is not a feasible extension and cannot be used in practice because we may not know how many Pareto optimal arms there are. Instead, we extend Auer A1 algorithm by replacing mean estimators with median estimators, adapting it to function as a median-based algorithm. We name this extension Auer-A1-M. This adjustment makes Auer-A1 more suitable for contaminated scenarios. We repeated the experiments using the same setups as presented in Section 6 of the paper, with the only difference being that the Auer-A1-M algorithm now utilizes medians instead of means. In the revised document, we present results over 100 iterations.
>
> # Answer to Requested Change 3
>
> In the revised manuscript, we have added an explanation of the limitations of previous work (Altschuler et al. 2019) to Section 1.1, highlighting the challenges their approach faces in the context of extending to PSI. Additionally, we provide an extensive overview of the key theoretical and algorithmic contributions of our work at the beginning of Section 5.2, clearly outlining the novel aspects of our analysis/method.

---

### Review · Reviewer_dMyo · 2024-09-03

**Summary Of Contributions:**

This paper studied a Pareto Set Identification Problem in multi-objective multi-armed bandits problem with contaminated reward. The paper proposed a sample-median based adaptive elimination algorithm and provided corresponding sample complexity bounds. Finally, the paper conducted numerical experiments on several different settings and compare their designed algorithm with several existing sample-mean based benchmarks.

**Audience:**

No

**Claims And Evidence:**

Yes

**Requested Changes:**

See weakness above.

**Strengths And Weaknesses:**

Strength:

1. It is interesting to see a sample-median based method.

Weakness and Requested Changes:

1. In Table 1, page 2, could the author provide the regret/sample complexity bound to help the audience better combine with the prior work?
2. It is not clear that why the author focused on median-based method rather than mean-based method? Could the authors provide more explanation?
3. The paper is not well-written.
4. There is no tightness / optimality check for the complexity bound. For example, are there any kinds of lower bounds for Theorem 1?
5. The paper only provides result in prescient adversarial corruption setting, what about oblivious adversary and malicious adversary?

---

> ### Author Response · Authors · 2024-09-27
> **Rebuttal**
>
> We thank the reviewer for their thoughtful comments and valuable insights.
>
> # Answer to Weakness 1
>
> We acknowledge the potential benefit of providing regret/sample complexity bounds in Table 1. However, many of these works have separate definitions and notations that have no counterparts in our work. Therefore, adding these bounds to the table would require defining and explaining foreign symbols, which may require further explanations on their respective contexts. For this reason we opted not to include regret/sample complexity bounds in Table 1. However, we provide (see Section 1 of the revised document) and discuss the sample complexity bound results for Altschuler et al. (2019) and Auer et al. (2016) in our work extensively, since they are the most related works to our work.
>
> # Answer to Weakness 2
>
> The choice of median-based estimators is crucial due to the nature of contamination in our setting, where contamination is unbounded in amplitude but bounded in frequency. Under this contamination model, the best contaminated mean is not a suitable approximation for the best true mean. When rewards are bounded within a large range, the contaminated mean can deviate from the true mean by a large amount. This can result in an error bound that is loose, even rendering it useless for prediction or estimation, especially when compared to the tighter bounds provided by our theoretical results. In other words, in the context of contamination, no matter how many samples are taken, mean estimators cannot reliably provide accurate estimations of the true mean values. In extreme cases, the contaminated distributions might not even possess finite first moments. In multi-objective scenarios, where each objective can be independently contaminated, the likelihood of encountering skewed data increases, making the mean statistic even less reliable. Thus, using the median ensures stability and accuracy in estimating arm performance under such challenging conditions.
>
> # Answer to Weakness 3
>
> We made an effort to adhere to the 12-page limit required by TMLR for a fast turnaround, which may have left some room for improvement in the writing. We tried our best to represent the contributions in a clear and concise manner in the revised version.

---

> ### Author Response · Authors · 2024-09-27
> **Rebuttal**
>
> # Answer to Weakness 4
> To address the question of tightness and optimality for the complexity upper bounds, we examined specific problem instances. Under these problem instances, the success conditions given in Definition 5 of our work implies the success conditions of the previous works. This further implies that our method (and also other Pareto accurate methods) is also bound to the lower bounds established in the mentioned work. A discussion of this analysis can be found in Remark 2 of the revised document, and we also give a detailed discussion below.
>
> To analyze the tightness of the upper bounds, we consider a specific problem instance where $M=1$. In this case, $P^*=\arg \max \_i m\_i$ and $\Delta\_i=\max \_j m\_j-m\_i$. Thus, the accuracy condition for Pareto accurate algorithms given in Definition 5 reduces to $\forall i \in P, m\_i \geq \max \_j m\_j-2 D-\alpha$. The coverage condition reduces to $\forall j \in P^*, \exists i \in P: m\_j-m\_i \leq 2 D$. Success conditions considered in Altschuler et al. (2019) require that any successful algorithm, for any $\alpha \geq 0$, should return a single arm $I$ such that $m\_{i^*}-m\_I-U\_{i^*}-U_I \leq \alpha$, where $i^*=\arg \max \_i m\_i$. In terms of our notation, this condition is $m\_I \geq \max \_j m\_j-2 D-\alpha$, which is equivalent to our accuracy condition. Thus, we can say that Altschuler et al. (2019) studies a success condition that is weaker than ours. Any lower bound for the success condition of Altschuler et al. (2019) also holds in our case. In particular, the lower bound stated in Theorem 18 of Altschuler et al. (2019) holds, which is given by $\Omega \left( \sum_{i \in [K] \setminus \{i^*\} } \frac{1}{{\max(\Delta}_i - 2D, \alpha)^2} \log (\frac{1}{\delta}) \right)$. In terms of dependence on $\alpha$, both Corollary 1 and 2 match this lower bound up to logarithmic terms.
>
> Let us consider another specific problem instance where the reward distributions are from family of distributions $\mathcal{F}\_{B, \bar{t}}$. Then, when $\epsilon =0$, we have $D = 0$ for all adversaries. Thus, the accuracy condition for Pareto accurate algorithms given in Definition 5 reduces to $\forall i \in P, \Delta\_i \leq \alpha$. The coverage condition reduces to $\forall j \in P^*, \exists i \in P: m\_j-m\_i \preceq 0$. However, by definition of a Pareto optimal arm $j$, such an arm $i$ cannot exist. Therefore, our coverage condition implies that all Pareto optimal arms must be in $P$, i.e., $P^* \subseteq P$. Thus, our coverage condition is equivalent to $P^* \subseteq P$. These two conditions are the same as success condition 1.a and 1.b of Auer et al. (2016), which is, $\forall i \in P: \Delta\_i^* \leq \alpha$ and $P^* \subseteq P$ respectively. Note that in Auer et al. (2016), $\Delta\_i^*=\max \_{j \in P^*} m(i, j)=\max (\{0, \min \_d(y\_j^d-y\_i^d)\})$ where $y\_i^d$ represents the mean as opposed to the median. Also, their Pareto set definition is in terms of means instead of medians. So their gaps are in terms of means, while our gaps are in terms of medians. However, when the distinction between using medians and means is ignored, our upper bounds match their lower bound up to logarithmic terms in terms of $\alpha$ dependency.
> # Answer to Weakness 5
>
> We would like to emphasize that our theoretical and experimental results explicitly cover all three types of adversaries: oblivious, prescient, and malicious. Our main theoretical results given in Theorem 1, Corollary 1 and Corollary 2 hold for all three adversaries. Our experimental results are presented across three datasets, each evaluated under a different adversarial setting. We conducted experiments on the SW-LLVM dataset under a malicious adversary setting, UVA/PADOVA diabetes dataset under an oblivious adversary setting and MovieLens dataset under a prescient adversary setting. We welcome any suggestions on additional details or adjustments to improve the clarity of our presentation.

---

### Review · Reviewer_4rFC · 2024-09-13

**Summary Of Contributions:**

This work proposed the R-PSI algorithm to identify the pareto set with contaminated bandit feedback. It discussed the three different types of attacks which are from oblivious adversary, prescient adversary and malicious adversary individually. It characterized the quality of an arm with its median. This work analyzed the sample complexity of the proposed R-PSI algorithm and also evaluated its performance with numerical experiments.

**Audience:**

Yes

**Claims And Evidence:**

No

**Requested Changes:**

Please refer to the part of **Weaknesses** in **Strengths And Weaknesses**.

Overall, I cannot see the analytical challenge and contribution upon comparing this work to Altschuler et al. (2019). Besides, some details are confusing to me and the author(s) may consider to polish some statements to ease reading.

**Strengths And Weaknesses:**

Strengths:
1. The comparison among settings considered in existing works and this work is clearly presented in Table 1.
1, The numerical experiments are run with various datasets including MovieLens, SW-LLVM, and that generated with UVA/PADOVA diabetes simulator.

Weaknesses:
1. The scenario of review bombing is not sufficiently convincing to imply the importance of this formulation. I'd appreciate more clarification of the motivation of this work.
1.  The explanation of 'Malicious adversary' in Section 3.1 is a bit confusing.
1. The materials presented in Sections 3.1 and 3.2 are similar to those in Altschuler et al. (2019). I suggest the author(s) to manifest the difference.
1. The work mentioned unbounded attacks in the beginning while Definition 5 seems to say that the accuracy will only be guarantee when the attacks are bounded, i.e., when $D<\infty$. I seek some clarification here.
1. The explanation of Algorithm 1 in Section 4 is a bit confusing. I suggest the author(s) to improve the description of the R-PSI algorithm and explain the sets more clearly.
1. The parameter $D$ does not appear in the bounds in Corollaries 1 and 2, or at least, the impact of $D$ on bounds in Corollaries 1 and 2 cannot be easily understand.
1. As Auer et al. (2016) studied the stochastic setting, I wonder how the result from Auer et al. (2016) can manifest the tightness of the bound in (8) as the contaminated feedback is considered in this work.
1. More explanation of theoretical results in Section 5 is appreciated, which may help manifest the contribution of this work.
1. Table 2 shows the comparison of the algorithms via different metrics. The implication of comparison via each metric can be explained in greater detail.

---

> ### Author Response · Authors · 2024-09-27
> **Rebuttal**
>
> We thank the reviewer for their thoughtful comments and valuable insights.
>
> # Answer to Weakness 1
>
> In many real-world applications—such as healthcare, finance, and resource allocation—decision-makers must balance multiple objectives while dealing with feedback that may be noisy and biased. Contamination in feedback can lead to suboptimal or unsafe decisions if not properly addressed. Our work provides a robust framework for efficiently identifying the Pareto optimal set even in the presence of corrupted feedback, which is essential for reliable multi-objective optimization in practical settings where data imperfections are inevitable.
>
> For example, in a cybersecurity setting, an attacker can perform a man-in-the-middle attack, where they intercept and modify communications between two parties. By selectively corrupting rewards from certain arms, the attacker can manipulate the algorithm's choices. As another example, in a clinical trial setting, contaminated feedback can arise from biased or manipulated patient reports. Suppose a MO-MAB algorithm is used to recommend treatment plans, where the objectives include maximizing efficacy, minimizing side effects, and reducing treatment cost. An adversary could influence the trial by injecting false feedback—e.g., by underreporting side effects or overreporting benefits of certain treatments. This could cause the algorithm to favor suboptimal treatments, putting patients at risk. Identifying the true Pareto-optimal treatments despite this contamination is critical to ensure safe and effective medical decision-making.
>
> # Answer to Weakness 2
>
> In the revised manuscript, we have updated the explanation of the malicious adversary in Section 3.1 to improve clarity. We have included an intuitive description of the concept, along with a concrete real-world example, specifically a man-in-the-middle attack, to help illustrate how such adversaries might operate in practice.
>
> # Answer to Weakness 3
>
> The materials introduced in Section 3.1 are extensions of contamination models from Altschuler et al., explaining how these models were adapted to the MO-MAB setting. To clarify exactly how they are extended and what changes there were, an explanatory sentence was added. Some of the materials given in Section 3.2 reviews the concentration results of Altschuler et al., which are, later in the work, used for the multi-objective setting. The works that are reviewed for this purpose were cited as Altschuler et al. (2019). Hence, their only difference is notation. Example 1 given in Section 3.2, however, is a novel material. Different than Altschuler et al. we also provide $R(\cdot)$ for subgaussian distributions which are widely used in bandits, thus extending the scope of the concentration theorems. Subgaussian distributions encompass a wide family of distributions commonly encountered in practice. This is important for improving the practical utility of the theoretical results, enabling their application to a wider array of real-world problems where reward distributions are often subgaussian.
>
> Indeed, our algorithm and its analysis have many technical differences than Altschuler et al., which can be found in Section 5.2 of the revised manuscript.
>
> # Answer to Weakness 4
>
> In the paper, we initially mention unbounded attacks in terms of value, meaning that there is no restriction on the magnitude of contamination in individual observations. In other words, when an arm $i$ is pulled and the observation in objective $d$ is contaminated (i.e. $B_{i, n}^d=1$), there is no bound on the observed contaminated value, $Z_{i, n}^d$. The unavoidable bias, $D := R(h_\epsilon)$, is how much these contamination may shift the median of the observed reward distribution $\tilde{Y}\_{i, n}^d$, from the median of the true reward distribution $Y_{i, n}^d$. Hence, it is not the attack amplitude. Moreover, $D$ is not dependent on the amplitudes (or bounds) of the contamination, it rather depends on the frequency of contamination $\epsilon$ and $R$ of the reward distribution. This is intuitive since the median statistic cannot be shifted much by a single outlier (which may not have a bound), but frequent outliers can shift the median statistic significantly. The use of $D$ in Definition 5 is due to the observations that if the median statistic of the observation is shifted (due to contamination) by the amount $D$, then the Pareto optimal arms cannot be distinguished from other 2D-optimal arms (no matter how many samples are taken). Therefore, it is impossible to approximate the optimal solution arbitrarily well. Hence, in Definition 5, the success conditions of coverage and accuracy account for this margin $D$. By doing so, the definition sets realistic expectations for the algorithm’s performance in adversarial settings, ensuring that it is not evaluated against unattainable levels of accuracy, and thus providing guarantees that are achievable in practice.

---

> ### Author Response · Authors · 2024-09-27
> **Rebuttal**
>
> # Answer to Weakness 5
>
> To enhance clarity, we have added a new figure (Figure 1) with a detailed caption to visually represent the phases of Algorithm 1 (R-PSI) in a simple case. Additionally, we have revised the explanation of the sets $O_1$ and $O_2$ to improve their clarity. We hope these changes provide a clearer understanding of Algorithm 1 and enhance overall interpretation.
>
> # Answer to Weakness 6
>
> To clarify, $D$ is defined as $D = R(h_\epsilon)$, meaning that $D$ is a function of $\epsilon$. It is important to note that in addition to $D$, there are other terms related to $\epsilon$ in the bounds, which arise from components not directly related to $D$. Separating $\epsilon$-dependent terms from $D$ can create a misleading impression about how the bounds scale with $\epsilon$. Therefore, we have chosen to express the bounds explicitly in terms of $\epsilon$, ensuring that the scaling behavior with respect to $\epsilon$ is accurately reflected. Moreover, even if we were to change the notation to $D_\epsilon$, the resulting bound might not accurately represent the dependency, as other $\epsilon$-dependent terms could alter the order of $D$ in the bound. This variability would make the order of $D$ changeable, which might cause confusion. Additionally, $D$ may also depend on other variables. For instance, $R$ in Example 1 depends on $\sigma$, and $R$ for distributions in $\mathcal{F}\_{B, \bar{t}}$, depends on $B m_2(F)$. The same issues regarding the separation of terms and scaling behavior exist for these variables as well. For these reasons, we opted not to present the bound in this form.
>
> # Answer to Weakness 7
>
> While our bound given in (8) includes additional terms due to contamination ($\epsilon$ and $\beta_{\bar{t}, \epsilon}$), and thus does not match the  results of Auer et al. (2016) when these terms are present, it retains a similar dependence on the accuracy parameter $\alpha$. The contamination-related terms are separate and affect the bound independently. This demonstrates that, while we account for contamination, the underlying structure of the bound (especially in terms of $\alpha$) remains closely aligned with that in Auer et al.'s (2016) work. We also analyzed how our upper bound relates to the adversary-free lower bound provided in Theorem 17 of Auer et al. (2016) for stochastic setting. This comparison highlights that, under the assumption of no contamination, the sample complexity of our method (R-PSI) aligns with this lower bound. This result is intuitive, as when $\epsilon = 0$, there is no contamination. Since $\epsilon$ is an input to our method, the concentration bounds used by R-PSI become tighter, allowing for faster elimination and covering decisions. Consequently, fewer samples are needed to classify points into optimal and suboptimal sets.
>
> To further see how the tightness of the upper bound in our method relates to the lower bound of Auer et al. (2016), we can also consider consider a specific problem instance where the reward distributions are from family of distributions $\mathcal{F}\_{B, \bar{t}}$. Then, when $\epsilon =0$, we have $D = 0$ for all adversaries. Thus, the accuracy condition for Pareto accurate algorithms given in Definition 5 reduces to $\forall i \in P, \Delta\_i \leq \alpha$. The coverage condition reduces to $\forall j \in P^*, \exists i \in P: m\_j-m\_i \preceq 0$. However, by definition of a Pareto optimal arm $j$, such an arm $i$ cannot exist. Therefore, our coverage condition implies that all Pareto optimal arms must be in $P$, i.e., $P^* \subseteq P$. Thus, our coverage condition is equivalent to $P^* \subseteq P$. These two conditions are the same as success condition 1.a and 1.b of Auer et al. (2016), which is, $\forall i \in P: \Delta\_i^* \leq \alpha$ and $P^* \subseteq P$ respectively. Note that in Auer et al. (2016), $\Delta\_i^*=\max \_{j \in P^*} m(i, j)=\max (\{0, \min \_d(y\_j^d-y\_i^d)\})$ where $y\_i^d$ represents the mean as opposed to the median. Also, their Pareto set definition is in terms of means instead of medians. So their gaps are in terms of means, while our gaps are in terms of medians. However, when the distinction between using medians and means is ignored, our upper bounds match their lower bound up to logarithmic terms in terms of $\alpha$ dependency.
> # Answer to Weakness 8
>
> In in the beginning of the Section 5.2 of the revised document, an overview and explanation of the theoretical results given in Section 5 is provided.
>
> # Answer to Weakness 9
>
> Section 6 has been updated to include implications of comparison via each metric in the revised document.

---

> ### Author Response · Authors · 2024-09-27
> **Rebuttal**
>
> # Answer to Requested Change
>
> While some of the concentration results and techniques bear resemblance to those used in prior work, they serve as necessary tools to address the core challenge of combining robustness against contamination with multi-objective optimization in bandit settings. This combination, while conceptually appealing, presents significant analytical hurdles. Methods presented in Altschuler et al. (2019) use a termination condition which cannot be straightforwardly extended to the multiobjective case (for more details on this, see the response to the Requested Change 2 of reviewer AJZo). R-PSI addresses these issues with intricate set operations. In Section 5.2 of the revised document, we provide an extensive overview of the key theoretical and algorithmic contributions of our work, clearly outlining the novel aspects of our analysis/method.
>
> We have made an effort to present the contributions clearly and concisely in the revised document. As mentioned before, we have also included a new figure (Figure 1) with a detailed caption to visually illustrate the phases of Algorithm 1 (R-PSI) in a straightforward case. We hope these changes provide a clearer understanding of Algorithm 1 and enhance overall interpretation.

---

### Decision · Action_Editor_DMPY · 2024-11-26

**Recommendation:** Reject

**Comment:**

As mentioned above, I doubt there would be any interested readers in this article as there are no new insights to be gleaned. In addition to reading the reviews and the rebuttals posted by the authors, I have perused the paper. The paper simply implements the key lemmas concerning median deviation from Altschuler et al. (2019) and extends them to the MO-MAB setting. There are no new  insights that one can learn about the multi-objective nature of the problem, as the authors simply consider particularizations to the M = 1 (1 objective setting) case. This is because the authors consider arms that are not correlated. One way forward is to explain what a reader can draw insights from about the multi-objective nature about the problem from the upper bound on the probability in Theorem 1. The authors are welcome to submit a major revision that clearly highlights the unique nature of the problem (even though it leverages tools from some papers like Altschuler et al. (2019)) in the results.

**Audience:**

Unfortunately, even though the authors consider a variant of the multi-armed bandit problem to the multi-objective case, I doubt there will be any interested readers as there are no new insights that would interest an audience. This is because the main theorem doesn't elucidate the complexities and intricacies of the multi-objective nature of the problem. Please see the detailed comments below.

**Claims And Evidence:**

The claims made in the paper are technically sound.

**Resubmission Of Major Revision:**

The authors may consider submitting a major revision at a later time.